# communications
# engineering

# End-to-end programmable computing systems

Yao Xiao [1], Guixiang Ma[2✉], Nesreen K. Ahmed [3], Mihai Capotă [2], Theodore L. Willke[2], Shahin Nazarian[1] & Paul Bogdan [1✉]

Recent technological advances have contributed to the rapid increase in algorithmic complexity of applications, ranging from signal processing to autonomous systems. To control this complexity and endow heterogeneous computing systems with autonomous programming and optimization capabilities, we propose a *unified, end-to-end, programmable graph representation learning* (PGL) framework that mines the complexity of high-level programs down to low-level virtual machine intermediate representation, extracts specific computational patterns, and predicts which code segments run best on a core in heterogeneous hardware. PGL extracts multifractal features from code graphs and exploits graph representation learning strategies for automatic parallelization and correct assignment to heterogeneous processors. The comprehensive evaluation of PGL on existing and emerging complex software demonstrates a 6.42x and 2.02x speedup compared to thread-based execution and state-of-the-art techniques, respectively. Our PGL framework leads to higher processing efficiency, which is crucial for future AI and high-performance computing applications such as autonomous vehicles and machine vision.

[1] University of Southern California, Los Angeles, CA 90089, USA. [2] Intel Labs, Hillsboro, OR 97124, USA. [3] Intel Labs, Santa Clara, CA 95054, USA. ✉email: guixiang.ma@intel.com; pbogdan@usc.edu

Many real-world applications across science and engineering (e.g., self-driving cars[1], digital signal processing[2], autonomous aerial[3], ground, and underwater systems[4–6]) urgently need increasing computational performance to match the rapid increase in the complexity of algorithms. Heterogeneous computing systems combine multiple types of hardware accelerators (e.g., graphics processing units GPUs, field-programmable gate arrays FPGAs) to achieve such computational gains.

To manage the need for computational gains, heterogeneous systems require intelligent, flexible, and efficient programming strategies that can match the requirements of real-world applications to the strengths of the heterogeneous architecture. To optimize this matching in terms of performance and energy efficiency, we need to improve the mappings, compiler transformations[7], accelerator utilization,[8] cache locality[9], and load balancing[10]. However, the existing monolithic programming models and task mapping to compute platforms do not fully exploit the recent heterogeneity as well as architectural innovations in current hardware systems. They also fail to efficiently use the heterogeneous processing elements which could exacerbate the load imbalance and communication inefficiencies[10–12]. For example, the conventional central processing unit CPU-only or GPU-only optimization techniques may not be suitable for a heterogeneous system that combines both. This is due to the architectural and programming model differences of these hardware accelerators. Therefore, novel optimization approaches are required to realize the potential of heterogeneous systems and achieve the goals of exascale performance.

Traditional compilation techniques rely on cost models (of relatively simple hardware) based on expert heuristics[13]. However, the growing need for heterogeneous hardware systems to improve performance has led to complexity increase in the hardware, and that in turn has also added to the complexity of the compilation targets. Thus, the traditional compilation techniques are insufficient to exploit the promising potential of heterogeneous hardware systems. For example, the search conducted with those techniques must be repeated for each new program and might require several compilations and executions. That makes them impractical for real-world applications[14]. Furthermore, due to workload imbalance, synchronization overhead, and resource sharing contention[10], the overall performance of those techniques may be sub-optimal.

Machine learning, in particular, deep-learning techniques[15], have been explored in compiler optimization to learn better cost models[15–18]. For example, a recent work[19] proposed an end-to-end deep reinforcement learning (DRL) method for ML compiler graph optimizations where the learned policies are generalized to new graphs and transferable to different tasks. Neurovectorizer[20,21] proposed an end-to-end deep reinforcement learning framework for the automatic vectorization of loops. In addition, ML-driven techniques are also used to optimize the execution time of tensor computation graphs[22] as well as deep neural networks in TASO[23] and SOAP[24]. However, there is still a need for compiler approaches that are capable of exploiting recent advances in machine learning to learn how to accurately map computations (e.g., kernels) onto heterogeneous hardware systems for a single application. Such techniques should be capable of learning better cost models in a dynamic and complex heterogeneous hardware systems under uncertain conditions that complicate the use of traditional compilation techniques. Moreover, such ML-driven techniques will help remove the burden of writing correct and efficient code from human programmers (particularly programmers with expertise outside of computer science).

To address these issues, we propose a machine learning framework to predict the optimal hardware device (e.g., CPU or GPU) to provide better performance given a software kernel, which is defined as the device mapping problem[25], as shown in Fig. 1. However, unlike the previous work[13,26,27] that uses ML to solve the device mapping problem, our approach focuses on how to accurately map computations onto heterogeneous hardware

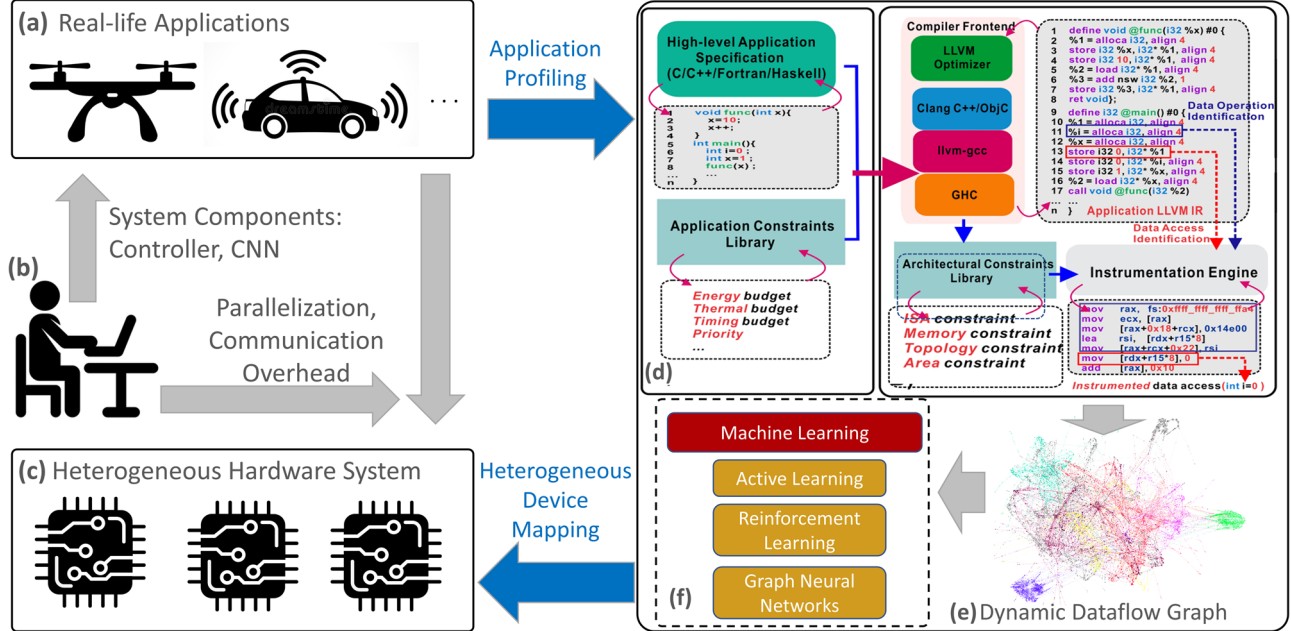

**Fig. 1 Autonomous heterogeneous computing system.** The recent advance of technologies enables the fast progress of autonomous cars and unmanned aerial vehicles (**a**). However, with the commonly used system components such as the controller and convolutional neural networks for image recognition (**b**), parallelization and communication overhead become inevitable concerns for programmers as the complicated and ever-changing software needs to be parallelized and executed on a heterogeneous system (**c**). The proposed framework makes the manual process autonomous without human intervention by profiling applications (**d**), constructing dynamic execution graphs (**e**), and mapping kernels onto the platform via machine learning models (**f**).

systems for a *single application*. As applications become more diverse and complex, it is inefficient to map them only onto one type of hardware accelerator. For example, in autonomous driving, the visualization and recognition tasks can be efficiently distributed, consisting of many *for* loops, onto cores in GPUs to provide higher parallelization. On the other hand, sequential decisions based on *if-else* statements require CPUs to provide fast execution on a single critical thread. In this example, GPUs provide a higher number of compute engines for parallel computing whereas CPUs have higher frequencies compared to GPUs, leading to faster execution of sequential threads. Therefore, a CPU/GPU heterogeneous system where the best features of both hardware devices are efficiently combined can achieve even further computational gains.

## Methods

**Setup**. Given a software program, our goal is to identify the *subgraphs* (i.e., code segments) that are optimal to run on CPUs or GPUs. Note that performance varies by use, configuration, and other factors. Learn more at www.Intel.com/PerformanceIndex. Our developed end-to-end framework consists of two components: a GAE and a GNN. The unsupervised learning model GAE is used to partition the complex program into several clusters/ kernels to be mapped onto heterogeneous systems. Supervised learning model GNN predicts the correct label for each kernel. In the implementation, we use kernels written in OpenCL[13] as training and testing data with 5-fold cross-validation for the GNN model. The ground-truth labels are either CPU or GPU for the kernels. In order to evaluate our proposed unified end-to-end programmable graph representation learning (PGL) framework, we first use the GAE model to partition the graphs, to find kernels suitable for either CPU or GPU. Next, different GNN models are used to predict the correct label to the underlying hardware. The configuration parameters of the heterogeneous system are listed below. The hardware contains 32 CPUs and 32 GPUs connected with the mesh-based network-on-chip. Each CPU has a 4-way 64KB L1 private cache, 256KB L2 shared cache, and 4GB memory, clocked at 2.4 GHz. Each GPU has 768MB memory with 86.4GB/s bandwidth, clocked at 575 MHz.

*Applications for the power-law relationship*. The power-law relationship between multifractal properties and system-level metrics can be characterized by analyzing 132 programs in 17 applications, which are discussed as follows: (1) Algebraic multigrid solver (AMS): the parallel algebraic multigrid solver for linear systems arising from problems on unstructured grids; (2) Fast sequence alignment (FSA): an ultrafast and memory-efficient tool for aligning sequencing reads to long reference sequences; (3) DNA sequence mapping (DSM): a software package for mapping DNA sequences against a large reference genome, such as the human genome, which consists of three algorithms: BWA-back-track, BWA-SW and BWA-MEM; (4) neural network (NN): an open source neural network framework written in C and CUDA; (5) Dijkstra (DA): Dijkstra shortest path; (6) Epidemic simulation (ES): a simulation of an epidemic, inspired by the 2019-20 novel Coronavirus Disease (COVID-19) pandemic; (7) Molecular dynamics (MD): a proxy application and research vehicle for particle code, in particular molecular dynamics; (8) Graph partitioning (GP): graph partitioning algorithms that contains multi-way partitioning algorithms, Fiduccia-Mattheyses-Sanchis (FMS), partitioning by locked moves (PLM), and partitioning by free moves (PFM); (9) Euler equation solver (EES): a mini-app that solves the time-dependent Euler equations of compressible gas dynamics in a moving Lagrangian frame using unstructured high-order finite element spatial discretization and explicit high-order

time-stepping; (10) Evolutionary algorithm (EA): Lamarckian evolutionary algorithm for molecular design and optimization; (11) IO proxy application (IPA): a multi-purpose, application-centric, scalable I/O proxy application for IO performance testing and multi-physics, HPC applications; (12) Mesh refinement application (MRA): an adaptive mesh refinement mini-app; (13) CNN: a convolutional neural network; (14) Poisson equation solver (PES): a solver for a standard Poisson equation using a conjugate gradient iteration with a simple or spectral element multigrid preconditioner on a block or linear geometry; (15) Monte Carlo kernel (MCK): a mini-app representing a key computational kernel of the Monte Carlo neutron transport algorithm; (16) HACC: a stand-alone version of hardware accelerated cosmology code (HACC)'s distributed-memory, pencil-decomposed, parallel 3D FFT; (17) Radiative transfer solver (RTS): a solver for the equation of radiative transfer in the multi-group two-moment approximation.

**Datasets**. We start by using the 256 heterogeneous device mapping OpenCL kernels[13] for the training and validation of GNNs. These kernels are labeled with CPU vs. GPU. We then manually convert these kernels to C code. Furthermore, we use standard application benchmarks to validate the overall PGL framework. These benchmarks are (1) Dijkstra to find the shortest path with an input of 100 nodes, (2) Fast Fourier transform with an input vector of size 4096, (3) K means clustering / partitioning with an input of 256 2D tuples, (4) Mandel to calculate the Mandelbrot set with an input of 4092 points; (5) Molecular dynamics with an input of 1024 particles, (6) Neural network with an input of 5 hidden fully connected layers, (7) Neurons with an input of 1024 neurons with the ReLU activation function, (8) Convolutional neural network with an input architecture of a convolutional layer connected with a max pooling layer and a fully connected neural network.

**Baseline comparisons**. When comparing the accuracy of the prediction results from GNN models, we use the following GNN models: (1) graph convolutional networks (GCN); (2) graph attention network (GAT); and (3) gated graph neural network (GGNN). We compare our graph representation to the Pro-GraML graph representation[26], NCC[28], and DeepTune[13], state-of-the-art techniques to represent programs as graphs. To quantify the benefits of graph partitioning, we compare the PGL framework with the following baselines in terms of the application performance: (1) K-means clustering connected with GCNs (KM+GCN); (2) hierarchical divisive clustering where all observations start in one cluster, and divisions are performed recursively as one moves down the hierarchy, connected with GCNs (HDC+GCN); (3) modularity-based community detection where an optimization model is proposed to measure the structure of graphs[10,29], connected with GCNs (MOD+GCN); (4) METIS graph partitioning[30] connected with GCNs (METIS +GCN); (5) feed-forward neural network, connected with GCNs[31] (NN+GCN). In addition, we compare the PGL framework in terms of the application performance with the following baselines: (1) threads in parallel programming (PAR); (2) modularity-based community detection to partition the graph into clusters and a heuristic mapping[10] (CommDet); (3) sliding window based neural network to locate specialized structures with a reinforcement learning based mapping (NN+RL)[31]; (4) Aladdin, a pre-RTL, power-performance simulator for fixed-function accelerators[32].

**Feature extraction**. Each node in a GNN is associated with numerous features, which are further used for clustering or

classification to make decisions at the node level or graph level. In the literature, the *code2vec*[33] and *inst2vec*[28] are commonly used to extract features by encoding programs via AST paths. However, the trained representations can put larger weights on names rather than code structure, which may lead to misclassification.

In order to exploit the graph structural information flow of programs, random walks reason about the number of adjacent nodes and the density of connections around a node[34]. A random walk is defined as a series of nodes, starting from $n_0$, the $j$th node is generated by the following distribution with a fixed length $l$.

$$P(n_j = j | n_i = i) = \begin{cases} \frac{w_{ij}}{\sum_j w_{ij}} & \text{if } (i,j) \in E \\ 0 & \text{otherwise} \end{cases} \quad (1)$$

where $w_{ij}$ is the edge weight between node $i$ and node $j$. In addition, multifractal analysis mathematically studies the structural complexity and topological heterogeneity of graphs[35]. The multifractal properties such as generalized fractal dimensions provide the higher-order statistics of a graph, which can be quantified by a finite box-covering method. That is, to study the different fractal structures in a graph, the box-covering method uses a box of the same size to cover the graph and then studies the relationship of the size of a box ($l$) and the number of nodes in the $i$th box of size $l$ ($N_i(l)$) as

$$\sum_i N_i(l)^q \sim l^{\tau(q)} \quad (2)$$

where $q$ is the distortion factor to differentiate the topological difference of fractal structures, and $\tau(q)$ is the mass exponent. Next, we can obtain the generalized fractal dimensions $D(q)$ from $\tau(q)$, which characterizes the different fractal structures of a graph.

$$D(q) = \frac{\tau(q)}{q - 1} \quad (3)$$

Therefore, to mine the local and scale-dependent topological properties of programs, we propose an algorithm in Supplementary Notes 1 that exploits random walks and multifractal concepts for encoding topological inter-dependencies (See the additional information for the full details of the algorithm.). Random walks explore the local topological density around node $i$ in a graph by finding random paths starting from node $i$ to node $j$. Once a random path is identified, we backtrack to the final destination node $j$ to find the subgraph $SG$ starting from $i$ to $j$. Next, we perform a multifractal analysis on the subgraph $SG$ to estimate its generalized fractal dimension. The time complexity of the algorithm is bounded by the Dijkstra strategy to find the shortest path for each node to every other node, which is $O(ElogV)$, where $E$ and $V$ are the numbers of edges and nodes, respectively. Finding all shortest paths in a graph has a time complexity of $O(EVlogV)$.

## Results

**Problem formulation and framework overview.** In order to combine the benefits of both CPUs and GPUs, as opposed to the traditional device mapping problem, we formulate a new problem to be considered within the high-performance computing and machine learning contexts: Given a complex software application, the goal is to learn a mapping function that predicts which code segments would run best on a specific hardware device in heterogeneous hardware platforms.

The scheduling and mapping of dataflow graphs are a well-studied research area including synchronous dataflow[36,37] and dynamic dataflow[38–40] extend the job-shop scheduling techniques to account for inter-processor communication costs. Pino et al.[41] show how to construct schedules for heterogeneous

multiprocessors. Falk et al.[42] give a parallel scheduling strategy based on clustering and demonstrate significant performance gains for multimedia applications. In recent work[16], most approaches to deep learning in compiler optimization borrow ideas from deep learning in natural language processing. However, the compiler domain has identified data structures such as abstract syntax trees and dataflow that exhibit the aspects more important for compiler optimization than the token sequences in natural language processing. Therefore, new graph representations of source code are developed to be used with the help of the recent advances in graph-based deep-learning models such as graph neural networks. For example[43], proposed a new compiler-based graph representation for deep-learning models of code. It incorporates the abstract syntax tree and control-data-flow graphs to understand program properties and enable deep learning such as graph neural networks on the graph properties. In addition, the concurrent execution of varying mixes of different applications on the many-core systems enables state-of-the-art research in predictable application execution in terms of run-time mapping. For example[44], proposed a hybrid mapping that achieves run-time predictability by combining the design-time analysis of application mappings with run-time management[45] provided a general, completely automated hybrid application mapping methodology for optimizing the mappings of multiple concurrent running soft real-time applications to a heterogeneous multiprocessor system on a chip to minimize latency and energy. However, previous work on graph representation of code fails to expose some interesting graph motifs in programming languages that are recurring at different scales. The proposed dynamic execution graph illustrates different self-repeating code structures that can be exploited in multifractal analysis to extract meaningful features.

Therefore, to decipher the complex higher-order inter-dependencies of real-world software, we represent their computations in programs (code) as a graph where each node represents a compute instruction and each edge represents an information flow from one instruction to another. While many prior works have employed machine learning methods from natural language processing to represent programs as a sequence of lexical tokens[13,46,47], recently there emerged a number of graph-based machine learning works that capture the structure of programs along with the syntactic and semantic information in the graph representation[28,33,43]. It has been observed that the graph-based representation learning strategies tend to have superior learning ability on the programs for many code analysis tasks, such as code similarity learning[48], program classification[49], etc. For instance[43], uses abstract syntax trees (ASTs) and control-dataflow graphs (CDFGs) independently to represent programs and apply GNNs for learning predictive compiler tasks on these graphs, which outperforms the recurrent neural networks (RNNs) on the token sequence representation of the programs. By modeling the program's control, data, and call dependencies as a graph[26], exemplified a GNN to learn representations from the graph for both node-level and graph-level tasks including compiler analysis, program classification, and device mapping. The graph representation of programs enables us to model the dynamic dependency structures of software programs and helps analyze program characteristics and automatically compile programs in heterogeneous platforms. The automation is achieved via graph-learning models to predict the type of each program from an initial feature matrix. In order to obtain the representative higher-order topological features from a graph representation, we perform a comprehensive multifractal analysis[35] and quantitatively relate the topological structures hidden in a software graph with computational performance on multiprocessor systems while accounting for communication and synchronization overheads.

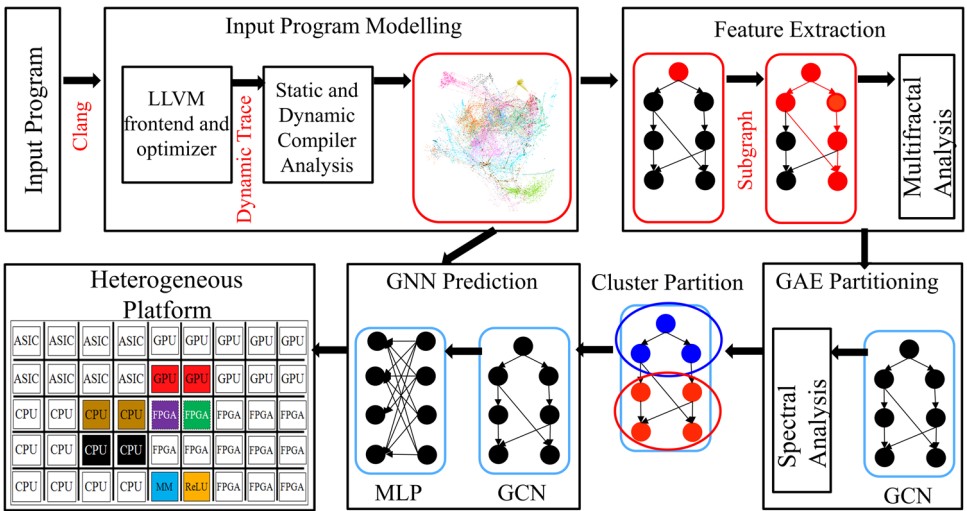

**Fig. 2 Overview of the proposed programmable graph-learning framework (PGL).** PGL constructs a dynamic execution graph for each input software program via low-level virtual machine (LLVM) intermediate representation (IR). PGL then utilizes a novel feature extraction algorithm based on random walks and multifractal analysis to construct node features that capture the topological dependencies and structures in dynamic execution graphs. These features are further used by a graph autoencoder (GAE) to partition the graph into clusters (i.e., software kernels) and a graph neural network (GNN) model such as graph convolutional networks (GCN) and multilayer perceptrons (MLP) to predict the best hardware device for each kernel.

To solve this challenging optimization problem, we propose a *unified, end-to-end, programmable graph representation learning* framework called PGL which is capable of mining the complexity of high-level programs down to the universal IR, extracting the specific computational patterns, and predicting which code segments run best on a specific core in heterogeneous hardware platforms. The proposed PGL framework, shown in Fig. 2, is flexible and capable of working with various graph representations of software code (e.g., regardless of the abstract syntax tree, or data-control flow graph). We also propose and evaluate a dynamic execution graph representation constructed from a partially executed trace of a code, where nodes represent low-level virtual machine (LLVM) intermediate representation (IR) instructions and edges represent control, data, and memory dependencies, which can better identify the structural information flow and capture memory dependencies.

**Dynamic dependency used in PGL is effective in representing code as graphs.** Recently, various graph representations were proposed for machine learning to represent and capture the latent information flow in a program (e.g., abstract syntax tree (AST)[33], contextual flow graph (XFG)[28], and control and dataflow graph (CDFG)[43]). These graph representations allow the compiler to analyze the effectiveness and correctness of programs, as well as enable parallel programming via graph partitioning in high-performance computing[10]. However, these statically compiled graphs have several limitations. First, memory dependencies are difficult to be identified. If not handled properly, this can exacerbate the data communication overhead and reduce the application performance. Second, the number of iterations in *for* and *while* loops cannot be statically determined. This plays a significant role in predicting whether the code is running in either CPU or GPU based on the workload. For example, if the number of iterations is small, it is ideal to run the code on CPU, because of the faster clock frequency. Otherwise, GPU is preferred because the number of cores on each chip is much denser to provide higher parallelism. Therefore, in order to overcome these drawbacks, we use the information generated from static compiler analysis and dynamic compilation to model the information flow in high-level programs as a dynamic execution graph. Next, we propose the following representation.

**Definition**. DYNAMIC EXECUTION GRAPH. A dynamic execution graph is a weighted directed acyclic graph $G = (V, E, W)$, where each node $v$, associated with an attribute $va$ indicating the type of the node (e.g., add, sub, store, or load), $(v, va) \in V$ represents an LLVM IR instruction; each edge $e$, associated with an attribute $ea$ indicating the type of dependencies (e.g., control, data, or memory), $(e, ea) \in E$ represents a dependency between two instructions; a weight $w \in W$ on each edge $e$ represents the amount of data communication between two instructions and the time to execute the instruction. It allows us to quantify communication overhead in the memory hierarchy with L1, L2, and L3 caches.

Note that the dataflow graphs in the literature are coarse-grained as each node represents a function in a program and each edge represents a signal path. However, each node in a dynamic execution graph introduced in this manuscript represents one LLVM IR instruction. It is coarse-grained enough to reduce simulation time and memory space for keeping track of all low-level assembly instructions and data structures. At the same time, It is fine-grained enough to express inter-dependencies between each pair of instructions dynamically collected.

The motivation for adopting a finer granularity analysis is three-fold: Firstly, the high-level languages and high-level programs may be designed in order to optimize certain software engineering objectives (e.g., modularity), but they are not taking advantage of or keep up with recent hardware innovations and developments (e.g., high parallelism in exascale computing). Secondly, the software development for certain applications may be done in a sub-optimal way without considering the time complexity in algorithms such as recursion used in Fibonacci numbers that leads to $O(2^N)$ where $N$ is the $N$th Fibonacci number. Thirdly, to bridge the gap between the high performance offered by heterogeneous hardware platforms and the high flexibility offered by general-purpose computing, we need a model of computation representation that allows us to flexibly capture the best of both worlds - the software and the hardware. Towards this end, we adopted the dynamic execution graphs with a finer-grain assembly code representation to retain the above-mentioned flexibility and provide higher software-hardware flexibility when compared to the dataflow graphs used in the literature. However, this finer granularity does not necessarily

mean higher communication overhead. Higher granularity means more nodes and more edges in our implementation but the communication overhead refers to the amount of communication that takes place between clusters after the partitioning. In order to prevent higher communication overhead, we introduce our partitioning algorithm that partitions the dynamic execution graphs into clusters. Indeed, we expect that a resulting cluster from the partitioning operation to be more similar to a node in the dataflow graphs in the literature. Each cluster is a sequence of instructions that are optimized to reduce data communication between clusters. Therefore, our graph representation with partitioning does not have a higher communication overhead. We optimize the inter-cluster communication to make sure the communication overhead between clusters is minimized.

To construct these dynamic execution graphs, we first collect the representative dynamic trace generated from executing a program. This trace contains a sequence of LLVM IR instructions to be executed. Then, for each instruction, we check if one of the following dependencies exists and insert a directed edge to construct the graph:

- *Data dependency*: Source registers of the current instructions depend on the destination registers of the previous instructions.
- *Control dependency*: Source registers of the function calls and branches depend on the destination register of the previous instructions.
- *Memory dependency*: Memory locations of the current store-load instructions are the same as the previous store-load instructions. We perform this memory alias analysis using "-basicaa -aa-eval -print-allalias-modref-info" in the LLVM environment.

Figure 3a–c shows some common zoomed-in graph patterns among dynamic execution graphs with high-level C code. Loops are commonly used in any programming language that can execute a group of statements multiple times. When arrays are used inside a loop statement, the corresponding dynamic execution graph has a star shape. The central node that is connected to different branches is the "getelementptr" LLVM IR. It is used to get the address of a sub-element of an aggregate data structure. Each branch corresponds to different instances of $a[i] = i$. When none of the arrays are used inside a loop, the corresponding dynamic execution graph has a mesh shape. When only sequential statements such as if-else are used in code, the corresponding dynamic execution graph has a tree shape to represent the information flow from the beginning to the end. Figure 3d–f shows the constructed code graphs for sequence alignment, signal processing, and convolutional neural networks, respectively. Note that a node is an LLVM IR instruction, not an operand or a high-level language (e.g., C/C++, Java) statement. Different from AST, XFG, and CDFGs, this specific graph representation in Fig. 3d–f makes explicit some hidden program information flows from the execution trace generated at run-time and analyzed via data, control, and memory dependencies. Each graph contains multiple fundamental graph patterns in (a), (b), and (c). For example, (d) clearly shows the mesh topology (b), and (e) has a star-shaped subgraph (a) that indicates the use of loops with arrays. In order to quantify the structural difference among the graphs, we also analyze the multifractal spectra of the graphs in (i), which validates that multifractal analysis is able to detect the topological structures in graphs. This helps us to design the feature extraction algorithm based on multifractal analysis in PGL.

In order to validate the effectiveness of PGL, we compare it with state-of-the-art techniques in terms of the accuracy of the prediction results on the same dataset[13]. We compare PGL

against the DeepTune and DeepLLVM using the code released by their authors. We also compare our graph representation against the ProGraML graph representation by extracting ProGraML graphs from the C versions of the kernels and training a GGNN on the graphs. Each dataset contains a set of kernels written in OpenCL and the labels associated with them. Each label is either 0 (CPU) or 1 (GPU). We then manually convert OpenCL into C in order to be used in ProGraML and PGL. We use 5-fold cross-validation to evaluate the machine learning models by partitioning each dataset into training, validation, and testing sets. Accuracy is measured by calculating the number of times that a framework is able to correctly predict the label for each kernel divided by the number of kernels. We repeat each experiment 100 times to report the mean and standard deviation. Precision is calculated by the true positive divided by the true positive plus the false positive. A recall is calculated by the true positive divided by the true positive plus the false negative.

As we can see from Table 1, PGL outperforms the state-of-the-art token-based DeepLLVM[47] by 1.03x and graph-based ProGraML[26] by 1.14x in terms of accuracy because it provides a novel way for program structural representation and enables the recent graph neural networks (GNNs) for the downstream tasks. In addition, we also test different graph neural networks including graph convolutional network (GCN)[50], graph attention network (GAT)[51], and gated graph neural network (GGNN)[52] along with PGL and it demonstrates that GGNN provides better accuracy compared to the rest by 1.04x.

In addition, we also test the impact of each framework on the fast convergence of the machine learning model in terms of accuracy for the NVIDIA (a) and AMD (b) datasets. More specifically, each machine learning is trained using 500 epochs to achieve stable results. In this experiment, we gradually remove 10 percent of training steps to understand which framework offers fast convergence in terms of accuracy. As we can see from Fig. 4, in general, PGL-GGNN offers the fastest convergence compared to others because it reaches the approximately optimal results at 60% whereas DeepLLVM reaches its optimal results at around 90%.

**The interdependence between advanced software code optimally executed on heterogeneous hardware exhibits a complex multifractal and universal behavior.** To decipher the mathematical relationship between the network properties (e.g., multifractal spectrum, generalized fractal dimension) and the system-level metrics such as the parallelization degree and communication overhead, we investigate different software kernels employed in high-performance computing and construct their corresponding code graphs. The code graphs exhibit a wide variety of self-similar structures due to loops and conditional statements. To quantify the higher-order topological complexity, we perform the multifractal analysis of code graphs and quantify their self-similar properties through the multifractal spectrum (Fig. 3g) and generalized fractal dimension (Fig. 3h). The width of the multifractal spectrum $f(\alpha)$ with respect to the Lipschitz-Holder exponents $\alpha$ measures the structural complexity and heterogeneity of a network[35]. Here, $\alpha$ quantifies the dimension of the fractal structure, and $f(\alpha)$ reflects the proportion of fractal structures with a given Lipschitz-Holder exponent $\alpha$, i.e., the distribution of fractal structures in the network. The multifractal spectrum of a monofractal graph is similar to a delta function where a single physical rule governs the graph structure at any scale and can be interpreted in terms of the system level as the fact that the graph can be mapped to either CPUs or GPUs. In contrast, the general multifractal spectrum exhibiting a non-zero width indicates that more than one physical rule governs the graph topology, which

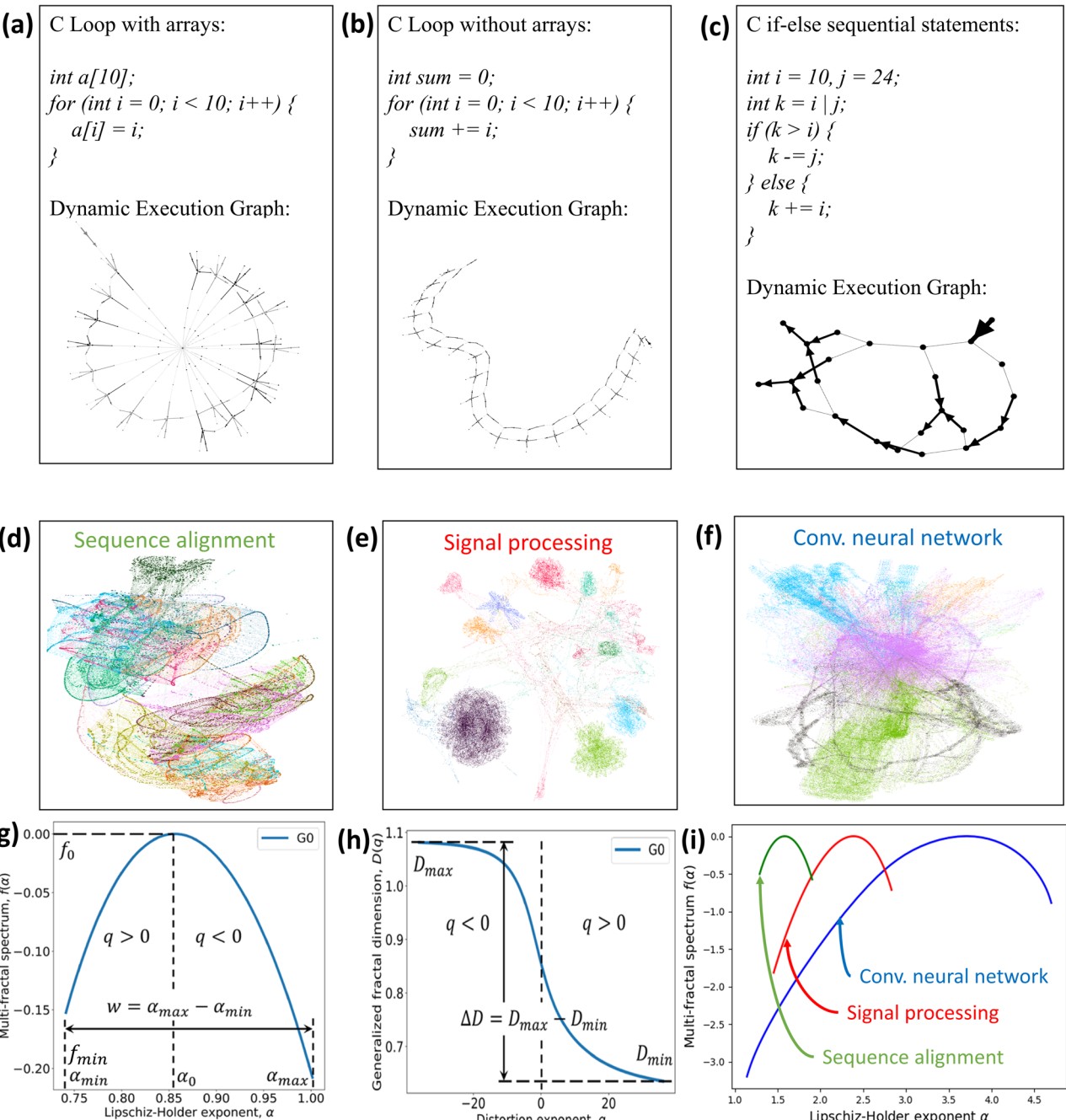

**Fig. 3 Dynamic execution graphs and multifractal properties.** Panel **a**, **b**, and **c** show basic graph patterns in graphs where the code contains either loops or sequential statements. Panel **d**, **e**, and **f** shows the constructed code graphs for sequence alignment, signal processing, and convolutional neural network, respectively. The graphs are a hybrid of fundamental graph patterns in (**a–c**). Panel **g** shows the multifractal spectrum and some definitions such as $\alpha_0$ and spectrum width $w$. Panel **h** shows a generalized fractal dimension for a graph. Panel **i** shows three multifractal spectra (green, red, and blue lines) for **d–f** to demonstrate multifractal spectrum can identify the heterogeneous graph structures in different dynamic execution graphs.

means that the graph is heterogeneous and should be carefully investigated in order to be mapped to both CPUs and GPUs.

For a dynamic execution graph constructed from a given software code implementation via compiler analysis, we partition it into several interdependent clusters to identify the optimal parallelization degree with respect to the characteristics of the heterogeneous computing system and minimize the inter-cluster weights (data communication overhead) via the optimization framework[10,53]. Each networked processing community represents a specific set of interdependent LLVM instructions, which is similar to a thread or process in operating systems. The inter-cluster weights represent the amount of data communication from one cluster to another, resulting from the optimization framework being minimized. To characterize the computational requirements and properties of various software codes, we consider two system-level metrics such as the parallelization degree and communication overhead. The parallelization degree is defined as the number of processing communities (clusters) generated from the optimization framework. The communication overhead is defined as the sum of inter-cluster weights between two clusters.

Dynamic execution graphs can exhibit some self-repeating patterns on different scales that we can exploit to capture and

understand the intrinsic graph structures. There are two fundamental techniques in programming languages: iteration and recursion. Iteration is a significant routine for a program to define a number of repetitions, usually via for-loops and while-loops. It corresponds to the mesh-like topology in graph representation. Recursion is the other major approach for a program to solve a problem where the solution depends on solutions to smaller instances of the same problem. It corresponds to a tree-like topology in graph representation. These two types of graphs can be analyzed at different scales to understand the recurring structures to extract the hidden features. For example, Fig. 5 shows an example of a for loop and its corresponding graph structure. In order to analyze its self-repeating patterns that can be seen in (b), we use the box-counting algorithm in multifractal analysis to calculate the dominant fractal dimension. In other words, we follow the definition of the measure to find the number of boxes ($N(B)$) with a box size $r$. The number of boxes is calculated by the optimal amount used to cover the entire graph. For example, when $r = 1$, the number of boxes $N(B)$ is the number of nodes in the graph, which is 116 in this case. When $r$ is the diameter of the graph, the number of boxes $N(B)$ is 1.

**Table 1 Comparison of the state-of-the-art techniques on the NVIDIA dataset (left) and AMD dataset (right).**

| Framework | Accuracy (%) | Precision | Recall | $F_1$ |
|---|---|---|---|---|
| DeepTune | 65.28 ± 5.32 | 0.68 | 0.68 | 0.68 |
| DeepLLVM | 88.64 ± 4.61 | 0.91 | 0.91 | 0.91 |
| NCC | 75.63 ± 4.85 | 0.80 | 0.80 | 0.80 |
| ProGraML-GGNN | 80.36 ± 4.19 | 0.83 | 0.83 | 0.83 |
| PGL-GCN | 87.66 ± 3.17 | 0.90 | 0.90 | 0.90 |
| PGL-GAT | 89.73 ± 3.88 | 0.92 | 0.92 | 0.92 |
| PGL-GGNN | 91.52 ± 3.14 | 0.94 | 0.94 | 0.94 |

| Framework | Accuracy (%) | Precision | Recall | $F_1$ |
|---|---|---|---|---|
| DeepTune | 68.4 ± 4.52 | 0.70 | 0.68 | 0.69 |
| DeepLLVM | 90.9 ± 2.14 | 0.93 | 0.93 | 0.93 |
| NCC | 78.5 ± 3.74 | 0.79 | 0.79 | 0.79 |
| ProGraML-GGNN | 86.6 ± 3.28 | 0.89 | 0.87 | 0.88 |
| PGL-GCN | 92.97 ± 2.79 | 0.93 | 0.93 | 0.93 |
| PGL-GAT | 93.36 ± 2.45 | 0.94 | 0.94 | 0.94 |
| PGL-GGNN | 93.87 ± 2.27 | 0.94 | 0.94 | 0.94 |

The $F_1$ score is the harmonic mean of the precision and recall.

We analyze 132 programs corresponding to 17 applications ranging from state-of-the-art high-performance solvers to the machine learning domain. Relying on dynamic and static compiler analysis, we transform each program into a dynamic execution graph and measure their corresponding multifractal properties and system-level metrics. Each dot in Fig. 6 represents one program. Supplementary Notes 2 discusses the general idea behind multifractal analysis. To investigate the existence of a mathematical relationship between the network properties and system-level computing metrics, we measure the generalized fractal dimension (Fig. 6a–b), the spectrum width (Fig. 6c, g), the spectrum height (Fig. 6d), the dominant Lipschitz-Holder exponent $\alpha_0$ (Fig. 6e, h), and the network complexity (Fig. 6f, i). We observe that the network and system-level computing metrics obey a power-law model (i.e., $ax^b$), indicating the existence of a universality phenomenon characterizing the efficient heterogeneous software-to-hardware optimization. For example, Fig. 6a shows the power-law trend between the generalized fractal dimension where $q = -10$ characterizing the rare network motifs and the parallelization degree. The higher this dimension, the more frequent the rare patterns in code graphs, and the higher the parallelization degree. Going beyond rare network motifs, we investigate the width of the multifractal spectrum which quantifies the richness in generating rules characterizing a dynamic complex software. Figure 6c, g shows the power-law relationship between the multifractal spectrum width and the parallelization degree, and the communication overhead, respectively, indicating a universality signature. The larger the multifractal spectrum width, the more heterogeneous the code graph and the higher the parallelization degree and communication overhead.

Once we analyze different graph properties from multifractal analysis such as generalized fractal dimension, spectrum height, and width, we are trying to relate the graph-level properties with some system-level metrics such as communication overhead and parallelization degree by fitting a power-law model into the data to help us understand the relationship. As we can see in Fig. 6, There exists such a model that can approximately estimate the system-level metrics from graph properties. This has two folds.

1. It provides a universal model that builds the relationship between the graph properties such as the multifractal spectrum and the system-level metrics such as the parallelization degree and communication overhead. If

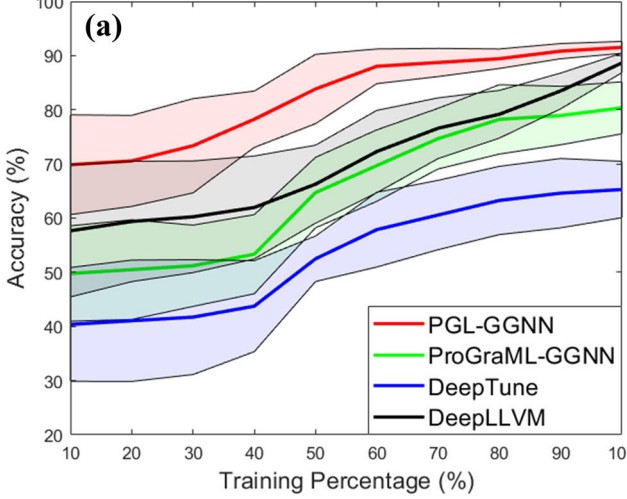
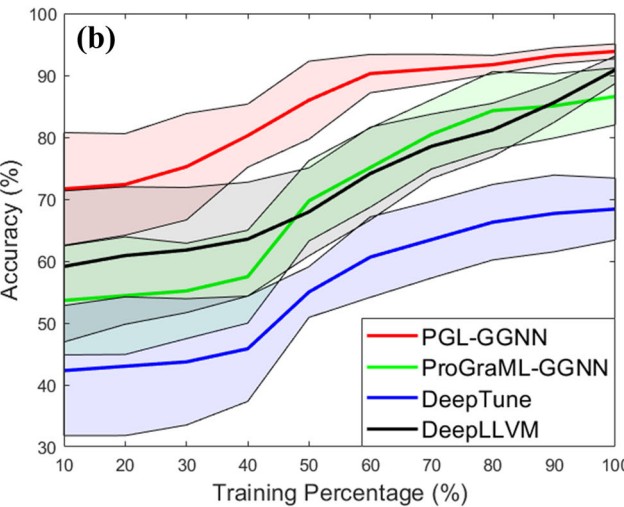

**Fig. 4 Framework comparison.** Convergence of normalized accuracy with different percentages of training steps in the NVIDIA (**a**) and AMD (**b**) datasets. Each color line indicates normalized accuracy for a given framework and each color shading associated with a line shows the standard deviation for the framework.

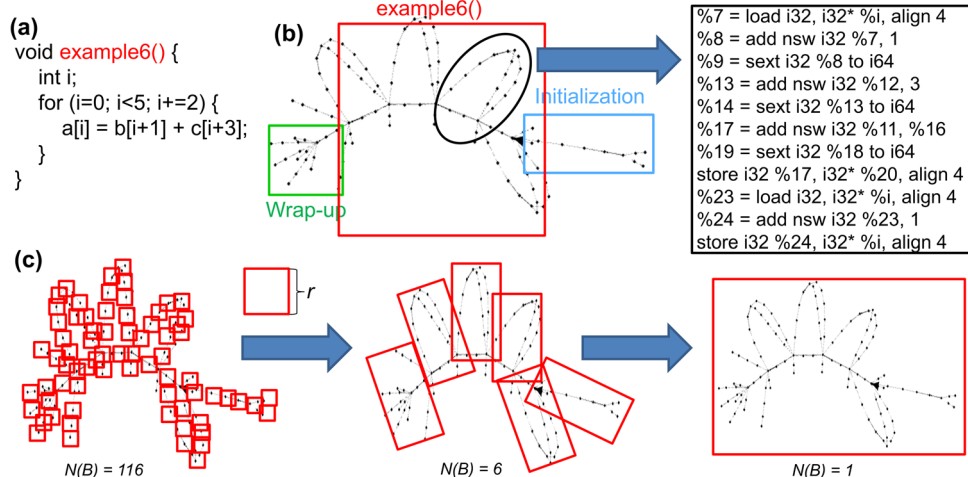

**Fig. 5 Example of code with its graph representation and box-counting algorithm used to analyze the multifractal properties. a** A loop kernel called example6 in red; **b** The dynamic execution graph with initialization in a blue rectangle and wrap-up in a green rectangle with a zoom-in view on one iteration of the loop; **c** The box-counting algorithm by varying the size of a box $r$ to count the number of boxes $N(B)$.

**Fig. 6 Multifractal analysis can characterize the universal power-law relationship between multifractal properties and system-level metrics.** Network multifractal properties are used as inputs to fit a power-law model $ax^b$ to find the relationship between network properties and system-level metrics. Panel **a–f** shows the parallelization degree of code graphs in terms of generalized fractal dimension (**a, b**), spectrum width (**c**), spectrum height (**d**), $\alpha_0$ (**e**), and complexity (**f**). Panel (**g–i**) shows the communication overhead for spectrum width (**g**), $\alpha_0$ (**h**), and complexity (**i**).

**Table 2 Comparison of different graph partitioning algorithms on the 17 applications**

|  | KM+GCN | HDC+GCN | MOD+GCN | METIS+GCN | NN+GCN | PGL-GGNN |
|---|---|---|---|---|---|---|
| Algebraic multigrid solver | 0.78 | 0.92 | 1.36 | 2.57 | 4.01 | 5.85 |
| Fast sequence alignment | 0.89 | 1.04 | 2.63 | 5.35 | 7.22 | 9.27 |
| DNA sequence mapping | 0.92 | 0.82 | 1.98 | 3.73 | 4.67 | 6.54 |
| Neural network | 0.86 | 1.12 | 4.52 | 8.42 | 9.64 | 11.85 |
| Dijkstra | 0.85 | 0.97 | 1.19 | 1.57 | 1.86 | 2.53 |
| Epidemic simulation | 0.98 | 1.21 | 2.52 | 4.22 | 6.53 | 8.34 |
| Molecular dynamics | 0.92 | 1.06 | 3.34 | 5.16 | 5.95 | 7.68 |
| Graph partitioning | 0.80 | 0.96 | 4.73 | 9.63 | 10.74 | 14.47 |
| Euler equation solver | 0.90 | 1.33 | 1.75 | 3.37 | 5.43 | 6.74 |
| Evolutionary algorithm | 0.94 | 0.89 | 1.42 | 2.56 | 4.12 | 6.2 |
| IO proxy application | 0.94 | 1.29 | 1.76 | 4.14 | 5.21 | 5.88 |
| Mesh refinement application | 0.93 | 1.05 | 2.78 | 3.75 | 4.52 | 6.32 |
| CNN | 0.88 | 1.27 | 2.56 | 5.12 | 6.43 | 7.69 |
| Poisson equation solver | 0.87 | 0.98 | 2.06 | 4.27 | 6.24 | 8.52 |
| Monte Carlo kernel | 0.79 | 0.87 | 1.89 | 3.64 | 4.88 | 6.03 |
| HACC | 0.92 | 0.86 | 2.21 | 4.83 | 5.75 | 7.84 |
| Radiative transfer solver | 0.97 | 1.26 | 2.44 | 5.12 | 6.70 | 8.93 |

**Table 3 Comparison of different frameworks on the 17 applications**

|  | PAR | CommDet | Aladdin | NN+RL | PGL-GGNN |
|---|---|---|---|---|---|
| 1. Algebraic multigrid solver | 1 | 1.32 | 2.04 | 1.65 | 5.85 |
| 2. Fast sequence alignment | 1 | 1.28 | 2.15 | 1.89 | 9.27 |
| 3. DNA sequence mapping | 1 | 1.46 | 1.96 | 2.21 | 6.54 |
| 4. Neural network | 1 | 1.88 | 3.21 | 2.67 | 11.85 |
| 5. Dijkstra | 1 | 1.22 | 1.35 | 1.05 | 2.53 |
| 6. Epidemic simulation | 1 | 1.09 | 1.77 | 2.04 | 8.34 |
| 7. Molecular dynamics | 1 | 1.15 | 2.20 | 1.6 | 7.68 |
| 8. Graph partitioning | 1 | 1.27 | 2.65 | 2.45 | 14.47 |
| 9. Euler equation solver | 1 | 1.33 | 2.85 | 2.5 | 6.74 |
| 10. Evolutionary algorithm | 1 | 1.54 | 2.54 | 2.2 | 6.2 |
| 11. IO proxy application | 1 | 1.32 | 2.96 | 2.56 | 5.88 |
| 12. Mesh refinement application | 1 | 1.65 | 2.33 | 2.75 | 6.32 |
| 13. CNN | 1 | 1.13 | 1.91 | 2.24 | 7.69 |
| 14. Poisson equation solver | 1 | 1.08 | 1.78 | 2.53 | 8.52 |
| 15. Monte Carlo kernel | 1 | 1.24 | 2.12 | 2.64 | 6.03 |
| 16. HACC | 1 | 1.35 | 2.52 | 2.31 | 7.84 |
| 17. Radiative transfer solver | 1 | 1.42 | 2.22 | 1.75 | 8.93 |

such a model can accurately capture the relationship, the optimal degree of parallelization can be calculated by the graph properties, without the manual tuning from a programmer. For example, if a dynamic execution graph from a piece of code has a spectrum width that equals 2, then we would expect communication overhead between 80 and $120 \times 10^8$ clock cycles, and the parallelization degree between 10 and 24. On the other hand, if a future platform can support millions of cores, then we can use this model to find how to write the code that can exploit the benefits in exascale computing.

2. It provides us with what design choices we can gain to develop the feature extraction algorithm. PGL contains a feature extraction algorithm used by the graph neural network to predict a label. It shows that multifractal analysis can capture the graph's topological structures, which can be further used in the feature extraction algorithm.

In addition, when the dataset is evaluated in the full-system simulation, we notice that PGL achieves 1.89x on average, which is consistently better compared to state-of-the-art graph partitioning algorithms such as METIS[30] and machine learning models, as

shown in Table 2. It can also provide 4.73x speedup on average, compared to state-of-the-art frameworks[31], as shown in Table 3.

### Graph auto-encoders can exploit network universality properties for partitioning large software into small kernels mapping them onto heterogeneous computing systems

*GAE-based partitioning of large software graphs into different kernels.* Graph auto-encoders (GAEs)[54] are a category of GNNs that aims at representing nodes into low-dimensional vectors in an unsupervised training fashion. They are different from other GNNs that are typically used for supervised or semi-supervised learning tasks. In our framework, the goal of the graph partitioning stage is to obtain a good partition for each LLVM graph based on a learned representation that captures the intrinsic structural information of the graph, such that the subgraphs preserve the inherent characteristics of the data, control, and memory dependencies in the LLVM graph. To this end, we propose a graph partitioning strategy based on the GAE[55] and spectral clustering[56] for our task, as shown in Supplementary Notes 3. Once the GAE partitions a dynamic execution graph into kernels, we further refine the partitions to minimize the communication overhead, which is discussed in Supplementary Notes 4.

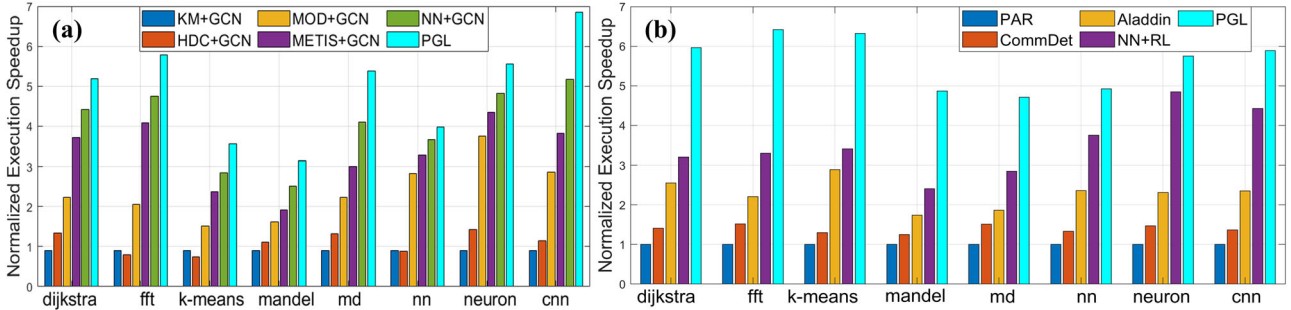

**Fig. 7 The breakdown of the execution time of each application in the standard dataset running on different frameworks.** The execution time, measured in clock cycles, is roughly divided into two parts: communication and computation in **a**. We also report communication overhead that is calculated by clock cycles in communication divided by the total clock cycles in **b**. As we can see, PGL, compared to the other frameworks, has the smallest communication overhead. It is because PGL has an optimization model that partitions the graph into different clusters to minimize inter-cluster communication.

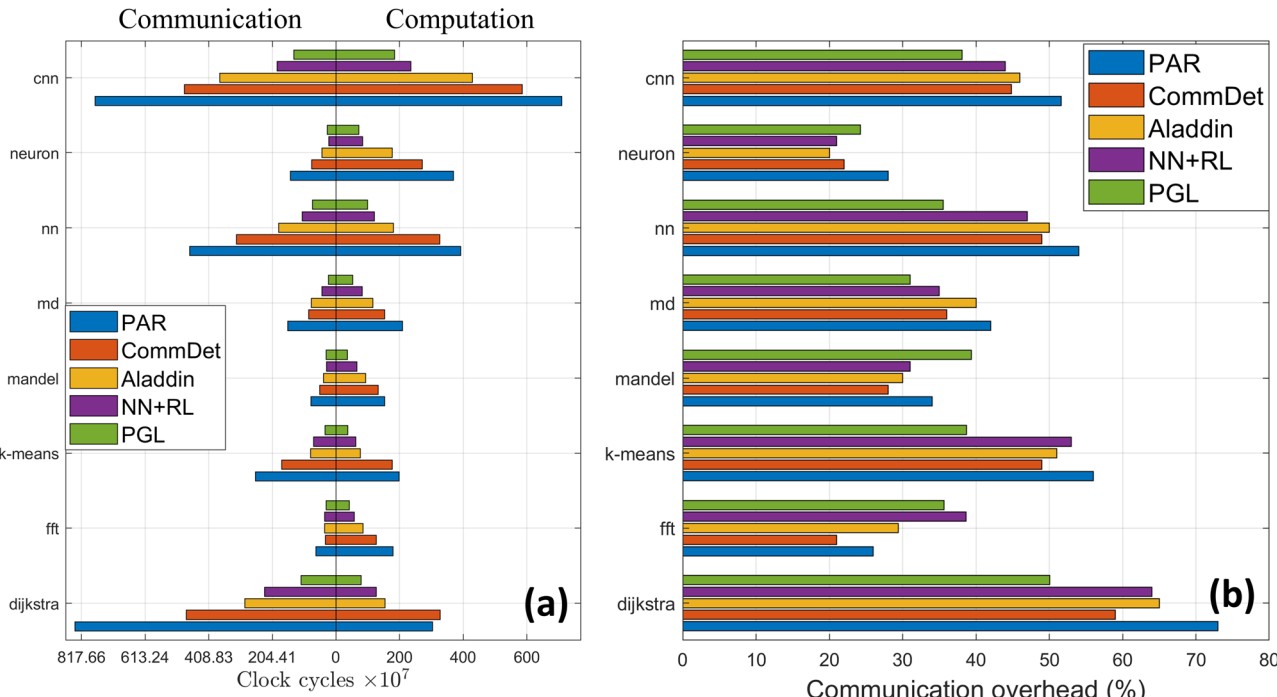

**Fig. 8 The breakdown of the execution time of each application in the real-life dataset running on different frameworks.** The execution time, measured in clock cycles, is roughly divided into two parts: communication and computation in **a**. We also report communication overhead that is calculated by clock cycles in communication divided by the total clock cycles in **b**.

At the partitioning level, it is true that we cannot guarantee a correct partition of a graph. This means node $i$ which should have been placed into cluster $i$ could be in cluster $j$. However, that does not necessarily lead to wrong results upon execution. The wrong results are usually caused by (1) missing instructions; (2) wrong order of instructions being executed; or (3) wrong data being fetched. However, none of the scenarios can happen in our construction of the dynamic execution graph thanks to the following safeguards: (1) Each graph contains all of the instructions and their direction dependencies, and the proposed partitioning does not remove the instructions and their dependencies. (2) The order of executing the instructions is preserved by exploiting our proposed topological sort strategy that guarantees directed dependencies among clusters, i.e., cluster $i$ is executed before cluster $j$ if there is a direct edge from cluster $i$ to $j$. (3) There are two possible cases when an instruction needs data: (i) whenever it loads data from memory and (ii) whenever it

depends on another instruction. When it needs data from memory, it can be in any cluster. For example, when an instruction from cluster $i$ needs data from another instruction that is in cluster $j$, the topological sort during the mapping stage resolves this situation by asking cluster $i$ to wait before the completion of cluster $j$ to make sure the data is available and sent to cluster $i$. To prevent livelock and deadlock situations, the optimization model used in partitioning has a constraint that prevents cyclic dependencies in clusters.

We performed experiments on two different benchmark suites in terms of application performance and reported the clock cycles spent either on communication or computation as shown in Figs. 7 and 8. For example, memory-intensive applications such as Dijkstra involve pointer address manipulation that requires frequent data fetch from memory. When running in PAR without any optimization, the communication overhead compared to the execution time is $820.52 \times 10^7$.

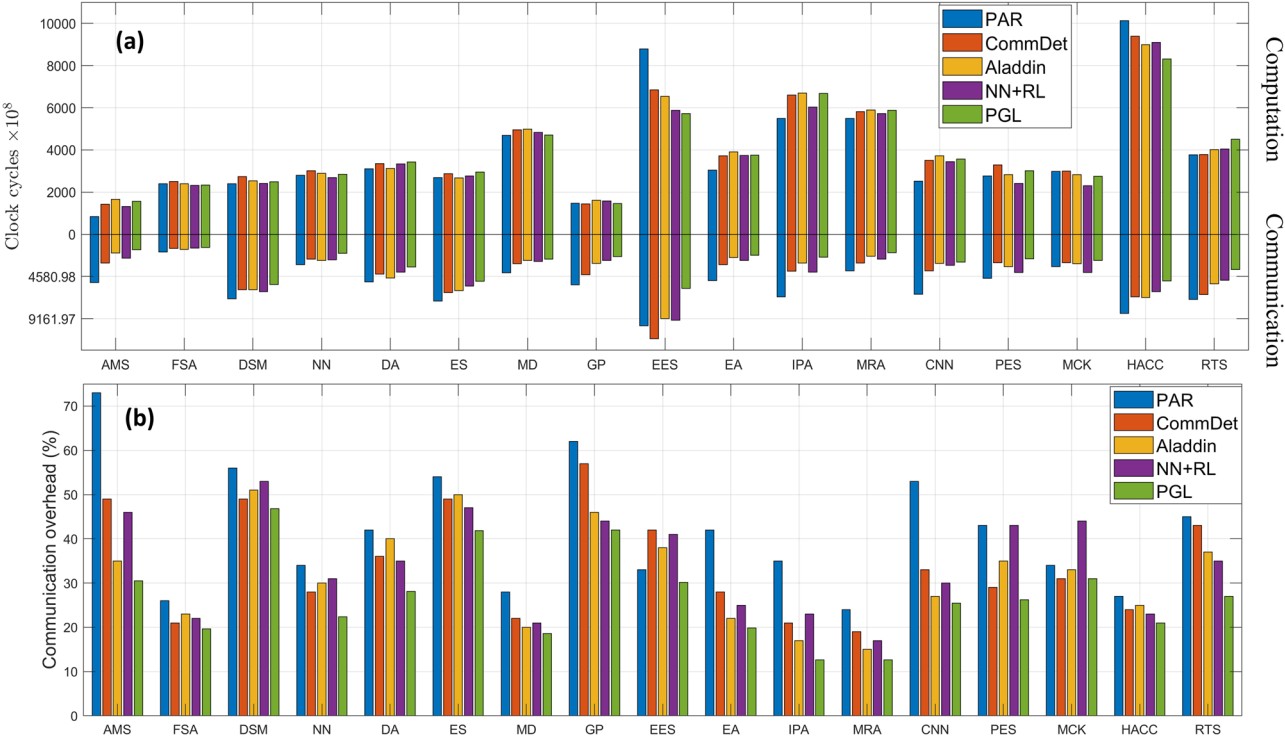

**Fig. 9 Experimental results.** Panel **a** shows comparison of different partitioning algorithms. We compare the graph partitioning GAE with different traditional algorithms. Panel **b** shows a comparison of different frameworks. We compare PGL with different frameworks in terms of application performance. We conclude that our approach can achieve 2.02x better compared to the state-of-the-art techniques.

However, PGL manages to reduce it to $109.38 \times 10^7$ via the optimization model to partition the graph into clusters while minimizing inter-cluster communication. For compute-intensive applications such as FFT, the communication overhead running in PAR is $63.18 \times 10^7$ whereas it is reduced to $31.10 \times 10^7$ in PGL. This indicates that even though PGL uses a finer-grained graph representation, the optimization-based partitioning approach would reduce the communication overhead between clusters.

*GNN-based mapping prediction on heterogeneous computing systems.* Once the kernels are further refined, next for each kernel, we use a GNN to predict the correct platform to execute the kernel by updating the node vectors iteratively in a similar fashion to the message passing. Note that our proposed PGL is a general framework that can leverage various GNN models for the device mapping prediction stage, whereas in this paper, we adopt three different variants of the GNN models: GCN[50], graph attention network (GAT)[51,57] and gated graph neural network (GGNN)[52], respectively. We also empirically investigate the comparative effectiveness of these GNN strategies in representation learning on the partitioned LLVM graphs for the graph classification task in heterogeneous device mapping.

We fix the graph neural network as GCN with two hidden layers and 32 neurons per layer, which is used to predict the correct label for each kernel. We compare the GAE with different partitioning algorithms such as *K*-means (KM), hierarchical divisive clustering (HDC), modularity-based community detection (MOD), METIS, and feed-forward neural network (NN) in terms of the total application execution speedup. As shown in Fig. 9a, for the partitioning models without machine learning such as KM, HDC, MOD, and METIS, the normalized execution speedup is smaller compared to the learning models such as NN and GAE. This is mainly because the kernels after graph partitioning are not well recognized by the GCN model. For the

learning models, GAE outperforms NN by up to 32% because the GAE takes into account the graph structures of code.

In order to validate the framework including the GAE and GNN models, we use the trained models to predict each application. As shown in Fig. 9b, we use the traditional thread-based parallel programming running on CPUs as our baseline and compare the PGL framework with community detection, a neural network with reinforcement learning, and Aladdin[32]. We observe that the PGL framework can provide up to 6.42x speedup compared to the baseline and 2.02x speedup higher compared to the state-of-the-art. Supplementary Notes 5–9 further show more experimental results on framework comparison.

## Discussion

We proposed PGL, an end-to-end learnable framework to predict which code segments run best on a specific hardware device. We first develop a node feature extraction algorithm based on random walks and multifractal analysis concepts to quantify the local structures of a program. We also measure different multifractal properties and find the universal relationship between those properties and system-level metrics such as parallelization degree and communication overhead. Next, we build the GAE together with a decoder and spectral clustering to find cluster partition from the distance matrix. Then, we use graph neural networks as the learning model to predict the type of each cluster. Our evaluation based on 32 CPUs and 32 GPUs confirms that the PGL framework can provide up to 6.42x speedup compared to the baseline and 2.02x higher speedup compared to the state-of-the-art technique.

We believe the universal power-law model between the multifractal properties and system-level metrics could serve as an indication to designers who plan to explore the best mapping for their applications. For example, if a designer wishes to know the optimal parallelization degree that is hard to find, the easiest approach could be to quickly collect a certain multifractal

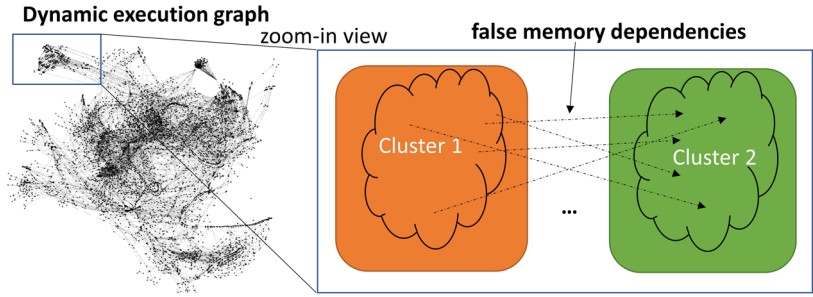

**(a) Few instructions -> not beneficial to parallelize**

**(b) Pointer manipulation -> random memory access -> false memory dependencies in alias analysis -> may increase the communication overhead**

**Fig. 10 PGL limitations. a** It is not beneficial for small code to pay for the overhead of PGL while mapping a few instructions onto cores. **b** Random memory accesses from pointer manipulation are not beneficial in PGL because there will be thousands of false memory dependencies due to LLVM alias analysis. This may increase the communication overhead.

property such as spectrum width or $\alpha_o$, and use the model to roughly determine the optimal degree. In the future, this learning framework is able to be generalized to any system that may include FPGA and emerging technologies.

In Fig. 10, we illustrate some of the limitations of PGL. We summarize the limitations and potential future extensions of the proposed framework as follows:

- First, the run-time profiling in the PGL framework only supports C and C++ code that involves complicated computation. Simple code with only a few lines is not beneficial in PGL (see Fig. 10a for a simple illustration example where a kernel contains three operations and it only has two clusters where one contains one instruction and the other contains two instructions after partitioning. The small code is not ideal in PGL because the overhead it spends on profiling and partitioning is not mitigated by mapping only a few instructions onto a specific core. In the future, developers could build more run-time systems that support different languages.
- Second, the PGL is not suitable for high-level programs that involve many memory random accesses, due to memory dependencies that are hard to identify (see Fig. 10b). As shown in Fig. 2b, we have a dynamic execution graph that involves memory address manipulation and indexing, which could lead to many false memory dependencies between clusters. While the LLVM alias analysis reports the MustAlias, MayAlias, and NoAlias dependencies, we treat MustAlias and MayAlias as memory dependencies and add an edge between two instructions irrespective of whether they are must or may alias. This may increase the communication overhead due to too many MayAlias cases. On one hand, after we partition the dynamic execution graph into clusters to minimize inter-cluster communication, if most of the memory dependencies are confined in one cluster, then it would not increase communication. On the other hand, if many false memory dependencies span across different clusters, then communication overhead gets

worse. In the future, this can be considered in the optimization model to partition the graph.
- Third, the high-level programs have to be compiled and run successfully to collect the execution trace that is required to build the dynamic execution graph, which could be time and space-consuming. In the future, developers could mitigate this issue by combining code run-time profiling and graph construction on the fly.

## Data availability
The authors declare that the data supporting the findings of this study are available within the paper and can be found in the "Results" section.

## Code availability
Code can be found in the GitHub repository https://github.com/xiaoyao0512/ProgrammableGraphLearning.

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

## Acknowledgements

P.B. and Y.X. gratefully acknowledge the support by the National Science Foundation under the Career Award CPS/CNS-1453860, the NSF award under Grant Numbers CCF-1837131, MCB-1936775, CMMI-1936624, and CNS-1932620, the U.S. Army Research Office (ARO) under Grant No. W911NF-23-1-0111, and the DARPA Young Faculty Award and DARPA Director Award under Grant Number N66001-17-1-4044, an Intel Faculty Award, and a Northrop Grumman grant. The funder had no role in study design, data collection and analysis, decision to publish, or preparation of the manuscript. The views, opinions, and/or findings contained in this article are those of the authors and should not be interpreted as representing official views or policies, either expressed or implied by the Defense Advanced Research Projects Agency, the Department of Defense, or the National Science Foundation.

## Author contributions

Y.X., G.M., N.K.A., M.C., T.L.W., S.N., and P.B. designed the research and methods. Y.X. and G.M. conducted the theoretical and experimental analysis of the methods. Y.X.,

G.M., N.K.A., T.L.W., S.N., and P.B. conceptualized research and provided advice for all parts of the work. Y.X., G.M., N.K.A., M.C., T.L.W., S.N., and P.B. provided tools. All authors wrote and reviewed the manuscript.

## Competing interests

The authors declare no competing interests.
