## [Peer Review File · Communications Engineering]

Reviewers' comments:

Reviewer #1 (Remarks to the Author):

The manuscript proposes a unified, end-to-end, programmable graph representation learning for code analysis and mapping on heterogeneous architectures.

The manuscript contributions comprise:

- A new method for code classification based on GNN
- An extension of the SoA dataset used for code classification with custom benchmarks and real-life applications.

While both contributions are relevant, the comparison with SoA is limited. Therefore, the authors should extend the comparisons, including recent development in the field and provide a more comprehensive analysis of the characteristics and statistical relevance of the new proposed benchmarks before considering the manuscript for publication.

More detailed comments:

1. Authors claim that the SoA comparison is limited to Graph-based methods only. However, to the best of the Reviewer's knowledge, the DeepTune approach is token-based. Moreover, the comparison is missing recently published works that, as DeepTune, are not graph-based but achieve better performance in code classification, which is in line with the one achieved with the proposed methodology.

Missing reference: Parisi et al., "Making the Most of Scarce Input Data in Deep Learning-Based Source Code Classification for Heterogeneous Device Mapping," TCAD 2022

2. DeepTune is well known to depend on the K-fold random split. Did the authors use the random seed fixed in the DeepTune released code or explore the stability of the result by repeating the experiment more times?

3. How are Table 1 results obtained? How is the dataset partitioned? How is the accuracy error computed? Which dataset has been used, NVIDIA GPUs or AMD GPUs or both?

4. It is unclear to the Reviewer how the newly added benchmarks compare with the previous ones in speedup distribution. Are there uniformly distributed?

5. Why did the authors not apply the SoA methodologies to the newly added benchmarks? Please add these comparisons.

Reviewer #2 (Remarks to the Author):

End-to-end Programmable Computing Systems

General Comments

This paper presents PGL, an end-to-end framework for partitioning and mapping of OpenCL programs to CPU/GPU systems, based on graph neural networks. Their method adds dynamic information to common static program representations. It uses this representation to partition a program into multiple kernels and find an optimal device mapping for this partition. The results of the performance prediction outperform state-of-the-art representations based only on static information, while the end-to-end framework outperforms other baselines the authors used for comparison.

Overall the idea of such an end-to-end framework is ambitious and valuable. The general approach seems like a good start. Unfortunately, present manuscript does not discuss several issues that are central to this problem, and consequently, it is unclear how valid the results are. Concretely, there are two distinct contributions here: the dynamic additions to performance estimation and the methods for partitioning. Both leave several open questions:

- The problem of automatic parallelization/partitioning has been well-studied and is notoriously difficult for languages with arbitrary aliasing, like C/OpenCL. The authors don't discuss how they address mitigate issues with aliasing and ensure the correctness of the partition.
- Given performance estimation for individual computational nodes, the mapping problem for data flow graphs is a well-studied research area. The authors don't acknowledge, discuss nor compare to established methods in this domain.
- It is unclear whether, how much or how the method considers the communication overhead for mapping multiple kernels to a dataflow system.
- The authors correctly observe that dynamic information is central to the CPU/GPU classification problem from [9, 22, 23, 27], but do not compare to other performance estimation methods using dynamic information (e.g. would the model beat LLVM-MCA on these unrolled dynamic execution graph? what about other more sophisticated performance estimation methods based on profiling?).

I believe the issue is partly with the format of this journal, as there is not enough space to discuss enough of the technical details. On the other hand, the authors spend much of the space discussing graphs on their parameters and analysis for concrete examples instead of explaining these issues above. For example Figures 3, 4 and 6 or the ablation study are all less interesting than how you ensure the correctness of your partition or a comparison with state-of-the-art methods in mapping or dynamic performance estimation (they do compare to the state-of-the-art in the graph representation).

I think this is promising work and an interesting idea, but without discussing the issues above, I cannot recommend publication.

Detailed Comments

Abstract:

- what is "the" universal IR?

Introduction:

- Are algorithms really becoming more complex? Perhaps the requirements, or the size of systems, but the algorithms themselves I'm not sure.
- Why such an emphasis on load balancing? Load balancing plays a role in achieving high performance or energy efficiency, as do many other things: good mappings, ILP, compilation, algorithmic improvements/overheads, efficient use of accelerators, good cache behavior/locality, etc, all arguably play at least as big a role as load balancing, if not bigger.
- Figure 1: Are frequent system and firmware updates really the main problem? I would strongly contend that. Whether these things make our lives better is debatable, and this debate is certainly outside the scope of this paper.
- "Formal" has a well-defined meaning in computer science (using a formal language/methods); the definition of the device mapping problem you give/cite there is not formal.
- The description is a mischaracterization of the state of research. Applications are distinct from computational kernels, and the mapping problem for (dataflow) applications is a well-established and studied method, as described in the general comments above. A useful, albeit dated survey can be found in "Mapping on multi/many-core systems: survey of current and emerging trends" by Kumar Singh, Shafique, Kumar and Henkel in DAC'13. Since then, several groups have continued to iterate on this problem and advanced the state of the art, e.g. the groups of J. Teich, J. Castrillon, A. Pimentel, M. Glass, among others.

Results:

- The authors discuss recent developments in code representation for machine learning without making clear that this refers specifically to this domain.
- While the plural of code, codes, is perfectly grammatical, it is not usually used to refer to multiple pieces of software code. Codes gives the impression of there being different encodings, like when people talk about different error-correcting codes, where the code (i.e. encoding) itself changes, and not the content of the coded artifact.
- That dataflow can be used to partition software is known since the 1970s and was well studied also in the 1980s. It is not clear, however, how the partitioning presented deals with issues of correctness due to the partition.
- Dynamic dataflow graphs have an established meaning that is used in a whole field of research. For a standard reference discussing dynamic dataflow alongside other dataflow models, see for example "Ptolemaeus, Claudius, ed. System design, modeling, and simulation: using Ptolemy II. Vol. 1. Berkeley: Ptolemy. org, 2014."
- Figure 3: the graphs a) b) and c) are nice visualizations; however, it is unclear what insight we can gain from such a visualization at this point. If it is the fractal properties, this is not well explained here, nor is the algorithm used to visualize them. Does the fractal property depend on this visualization algorithm? A closer (zoomed in) view showing the actual labels for nodes and edges of the graph would be much more insightful, as it would give an idea of the actual structure of the graph.
- Figure 4: is this really necessary? I did not find the graph very insightful, it shows some linear regressions for some particular parameters on the model for some examples. More generally,

Multifractal analysis is not a well-established method that the authors can expect a reader to know of, nor are Lipschitz-Holder exponents, etc. While the authors do explain some of these concepts, they do not really explain what the point is of showing these details of the modeling. It seems to be an unnecessary level of detail, given other omissions in the paper (like the runtime system for the partitioned graphs, the discussion and comparison of the mapping literature, etc.)

- The analysis of the 132 programs and description of their fractal properties seems to be interesting, but again, I did not understand the point of it (see above). What insight does this give me into end-to-end compilation in general, the overall goal of the paper?

Methods:

- The language in the methods section is clearly distinct from the language before, showing that less care was taken for the proof reading (e.g. several articles are missing before nouns, etc)

- Figure 6: this three-dimensional plot is hard to read. In general it is hard to see three-dimensional visualizations in a two-dimensional sheet of paper. I would recommend showing the two projections individually instead.

- The comparison to state of the art is confusing: how is gem5-aladdin, a simulator, a comparison to PGL? More importantly, how is this, and a generic neural network with reinforcement learning state-of-the-art? It certainly is not state of the art in mapping (see above), which explains how the numbers of the speedup can be so high!

- While the ablation study and discussion on parameter tuning is certainly interesting, it is again not the level of detail I would expect in this article, in particular with regards to the omissions discussed above.

Response to Referees

March 16, 2023

We thank all the reviewers for their feedback. We include below our response letter to the referees. The comments from each reviewer are highlighted in bold, our responses to the reviewers are discussed after each question/comment, and the modifications we made in the revised manuscript are highlighted in blue.

1 Reviewer #1

We thank the reviewer for the suggestions. We have summarized the major modifications in the revised manuscript based on your comments and answered each comment as follows.

Major modifications:

- We have added DeepLLVM [PBBA21, dee] as a comparison using both NVIDIA and AMD datasets in Table 1 and 2.
- We have performed the evaluation on the listed 17 applications on different graph partitioning algorithms and frameworks in Table 5 and 6.

1. **Authors claim that the SoA comparison is limited to Graph-based methods only. However, to the best of the Reviewer’s knowledge, the DeepTune approach is token-based. Moreover, the comparison is missing recently published works that, as DeepTune, are not graph-based but achieve better performance in code classification, which is in line with the one achieved with the proposed methodology. Missing reference: Parisi et al., “Making the Most of Scarce Input Data in Deep Learning-Based Source Code Classification for Heterogeneous Device Mapping,” TCAD 2022.**

Indeed, DeepTune is token-based and we compare our graph-based approach with it and another SoA graph-based approach called ProGraML as shown in Table 1 to validate that our approach is better than the state-of-the-art techniques in the field. We have added the reference in the Results section, which is reproduced below. We also evaluated it in all of the experiments.

While many prior works have employed machine learning methods from natural language processing to represent programs as a sequence of lexical tokens [NNPN18, CPWL17, PBBA21], ...

2. **DeepTune is well known to depend on the K-fold random split. Did the authors use the random seed fixed in the DeepTune released code or explore the stability of the result by repeating the experiment more times?**

We used the code originally released by DeepTune. Nevertheless, we have repeated all of the experiments 100 times and updated the numbers in the revised manuscript and the following table in the response of Q3.

3. **a. How are Table 1 results obtained? b. How is the dataset partitioned? c. How is the accuracy error computed? d. Which dataset has been used, NVIDIA GPUs or AMD GPUs or both?**

We have added more descriptions on discussing the updated Table 1, which is reproduced below.

Table 1: Comparison of the state-of-the-art techniques on the NVIDIA dataset.

	Accuracy (%)	Precision	Recall	F₁
DeepTune	65.28 ± 5.32	0.68	0.68	0.68
DeepLLVM	88.64 ± 4.61	0.91	0.91	0.91
NCC	75.63 ± 4.85	0.80	0.80	0.80
ProGraML-GGNN	80.36 ± 4.19	0.83	0.83	0.83
PGL-GCN	87.66 ± 3.17	0.90	0.90	0.90
PGL-GAT	89.73 ± 3.88	0.92	0.92	0.92
PGL-GGNN	91.52 ± 3.14	0.94	0.94	0.94

Table 2: Comparison of the state-of-the-art techniques on the AMD dataset.

Framework	Accuracy (%)	Precision	Recall	F₁
DeepTune	68.4 ± 4.52	0.70	0.68	0.69
DeepLLVM	90.9 ± 2.14	0.93	0.93	0.93
NCC	78.5 ± 3.74	0.79	0.79	0.79
ProGraML-GGNN	86.6 ± 3.28	0.89	0.87	0.88
PGL-GCN	92.97 ± 2.79	0.93	0.93	0.93
PGL-GAT	93.36 ± 2.45	0.94	0.94	0.94
PGL-GGNN	93.87 ± 2.27	0.94	0.94	0.94

We are using the NVIDIA dataset as mentioned in the original manuscript in the method section on page 8. Nevertheless, we are also providing the results for the AMD dataset at the end of this comment. Each dataset contains a set of kernels written in OpenCL and the labels associated with them. Each label is either 0 (CPU) or 1 (GPU). We then manually convert OpenCL into C in order to be used in ProGraML and PGL. We use 5-fold cross validation to evaluate the machine learning models by partitioning each dataset into training, validation, and testing sets. Accuracy is measured by calculating the number of times that a framework is able to correctly predict the label for each kernel divided by the number of kernels. We repeat each experiment 100 times to report the mean and standard deviation. Precision is calculated by the true positive divided by the true positive plus the false positive. A recall is calculated by the true positive divided by the true positive plus false negative. A F1 score is calculated as follows, which is needed when we want to seek a balance between precision and recall.

$$F1 = 2 \times \frac{Precision \times Recall}{Precision + Recall}$$

We compare our approach with different state-of-the-art token-based or graph-based frameworks to validate our approach. Each framework takes as inputs each kernel with some meta-information that describes the kernel and outputs either 0 or 1 indicating either CPU or GPU. DeepTune is a token-based approach that [CPWL17, cum] develops a deep neural network that learns heuristics over code, entirely without using code features. The neural network constructs code representations and learns how to optimize for the best performance without the manual labor work. DeepLLVM [PBBA21, dee] is another token-based approach that exploits the meta-information in the dataset through Siamese networks to increase the dataset cardinality. Neural code comprehension (NCC) [BNJH18a, ncc] is a general machine learning technique to learn semantics from raw code that can be used for downstream tasks such as the device mapping problem. ProGraML [CFBN⁺21, pgc] is a graph-based program representation for data flow analysis and compiler optimizations. Similar to this work, it transforms an input program into the corresponding graph representation that maintains dependencies in the code. Unlike ProGraML, PGL graph representation involves dynamic compilation that enables the novel topology in the information flow to differentiate different programs. As we can see from Table 1, PGL outperforms the state-of-the-art token-based DeepLLVM by 1.03x and graph-based ProGraML by 1.14x in terms of accuracy because it provides a novel way for program structural representation and enables the recent graph neural networks (GNNs) for the downstream tasks. In addition, we also test different graph neural networks including graph convolutional network (GCN) [KW16], graph attention network (GAT) [VCC⁺17], and gated graph neural network (GGNN) [LTBZ15]

Figure 1: Convergence of normalized accuracy with different percentages of training steps in the NVIDIA (a) and AMD (b) datasets.

along with PGL and it demonstrates that GGNN provides better accuracy compared to the rest by 1.04x.

In addition, we also test the impact of each framework on the fast convergence of the machine learning model in terms of accuracy for the NVIDIA (a) and AMD (b) datasets. More specifically, each machine learning is trained using 500 epochs to achieve stable results. In this experiment, we gradually remove 10 percent of training steps to understand which framework offers fast convergence in terms of accuracy. As we can see from Figure 1, in general, PGL-GGNN offers the fastest convergence compared to others because it reaches approximately optimal results at 60% whereas DeepLLVM reaches its optimal results at around 90%.

4. **It is unclear to the Reviewer how the newly added benchmarks compare with the previous ones in speedup distribution. Are there uniformly distributed? Why did the authors not apply the SoA methodologies to the newly added benchmarks? Please add these comparisons.**

As suggested, we have applied PGL to the newly added benchmarks and updated the manuscript, which is reproduced below.

We have collected 132 kernels in 17 applications, which are described as follows in Table 3. (1) Algebraic multigrid solver: the parallel algebraic multigrid solver for linear systems arising from problems on unstructured grids; (2) Fast sequence alignment: an ultrafast and memory-efficient tool for aligning sequencing reads to long reference sequences; (3) DNA sequence mapping: a software package for mapping DNA sequences against a large reference genome, such as the human genome, which consists of three algorithms: BWA-backtrack, BWA-SW and BWA-MEM; (4) Neural network: an open source neural network framework written in C and CUDA; (5) Dijkstra: Dijkstra shortest path; (6) Epidemic simulation: a simulation of an epidemic, inspired by the 2019-20 novel Coronavirus Disease (COVID-19) pandemic; (7) Molecular dynamics: a proxy application and research vehicle for particle code, in particular, molecular dynamics; (8) Graph partitioning: graph partitioning algorithms that include contains multi-way partitioning algorithms, Fiduccia-Mattheyses-Sanchis (FMS), partitioning by locked moves (PLM), and partitioning by free moves (PFM); (9) Euler equation solver: a mini-app that solves the time-dependent Euler equations of compressible gas dynamics in a moving Lagrangian frame using unstructured high-order finite element spatial discretization and explicit high-order time-stepping; (10) Evolutionary algorithm: Lamarckian evolutionary algorithm for molecular design and optimization; (11) IO proxy application: a multi-purpose, application-centric, scalable I/O proxy application for IO performance testing and multi-physics, HPC applications; (12) Mesh refinement application: an adaptive mesh refinement mini-app; (13) CNN: a convolutional neural network; (14) Poisson equation solver: a solver for a standard Poisson equation using a conjugate gradient iteration with a simple or spectral element multigrid preconditioner on a block or linear

Table 3: Dataset summary

	Description	Source	Number of kernels
1	Algebraic multigrid solver	[amd]	8
2	Fast sequence alignment	[LTPS09, bow]	20
3	DNA sequence mapping	[Li13, bwa]	21
4	Neural network	[nn]	2
5	Dijkstra	[dij]	1
6	Epidemic simulation	[FLNG ⁺ 20, epi]	1
7	Molecular dynamics	[md]	2
8	Graph partitioning	[DA97, par]	17
9	Euler equation solver	[ees]	9
10	Evolutionary algorithm	[KDW22, evo]	4
11	IO proxy application	[iop]	6
12	Mesh refinement application	[amr]	16
13	CNN	[FYP ⁺ 21, cnn]	6
14	Poisson equation solver	[GML ⁺ 16, nek]	3
15	Monte Carlo kernel	[xsb]	3
16	HACC	[hac]	9
17	Radiative transfer solver	[ime]	4

Table 4: Configuration parameters

CPU	Cores	64 In-order cores, 16 MSHRs
	Clock frequency	2.4 GHz
	L1 private cache	128KB, 4-way associative 32-byte blocks
	L2 shared cache	2MB, distributed
	Memory	4 GB, 8 GB/s bandwidth
GPU	Number	512
	Clock Frequency	575 MHz
	Memory	768 MB, 86.4 GB/s bandwidth
Network	Topology	Mesh
	Routing algorithm	XY routing
	Flow control	Virtual channel flit-based

geometry; (15) Monte Carlo kernel: a mini-app representing a key computational kernel of the Monte Carlo neutron transport algorithm; (16) HACC: a stand-alone version of hardware accelerated cosmology code (HACC)’s distributed-memory, pencil-decomposed, parallel 3D FFT; (17) Radiative transfer solver: a solver for the equation of radiative transfer in the multi-group two-moment approximation.

As shown in Table 5 and 6, we use the full-system simulation in gem5 to obtain the speedup compared to different graph partitioning algorithms and frameworks when using the dataset in Table 3. The configuration of gem5 is listed in Table 4. We notice that PGL achieves 1.89x on average, which is consistently better compared to state-of-the-art graph partitioning algorithms such as METIS [LPS⁺15] and machine learning models. It can also provide 4.73x speedup on average, compared to state-of-the-art frameworks [XNB19].

Table 5: Comparison of different graph partitioning algorithms on the 17 applications

	KM+GCN	HDC+GCN	MOD+GCN	METIS+GCN	NN+GCN	PGL-GGNN
Algebraic multigrid solver	0.78	0.92	1.36	2.57	4.01	5.85
Fast sequence alignment	0.89	1.04	2.63	5.35	7.22	9.27
DNA sequence mapping	0.92	0.82	1.98	3.73	4.67	6.54
Neural network	0.86	1.12	4.52	8.42	9.64	11.85
Dijkstra	0.85	0.97	1.19	1.57	1.86	2.53
Epidemic simulation	0.98	1.21	2.52	4.22	6.53	8.34
Molecular dynamics	0.92	1.06	3.34	5.16	5.95	7.68
Graph partitioning	0.80	0.96	4.73	9.63	10.74	14.47
Euler equation solver	0.90	1.33	1.75	3.37	5.43	6.74
Evolutionary algorithm	0.94	0.89	1.42	2.56	4.12	6.2
IO proxy application	0.94	1.29	1.76	4.14	5.21	5.88
Mesh refinement application	0.93	1.05	2.78	3.75	4.52	6.32
CNN	0.88	1.27	2.56	5.12	6.43	7.69
Poisson equation solver	0.87	0.98	2.06	4.27	6.24	8.52
Monte Carlo kernel	0.79	0.87	1.89	3.64	4.88	6.03
HACC	0.92	0.86	2.21	4.83	5.75	7.84
Radiative transfer solver	0.97	1.26	2.44	5.12	6.70	8.93

Table 6: Comparison of different frameworks on the 17 applications

	PAR	CommDet	gem5-aladdin	NN+RL	PGL-GGNN
Algebraic multigrid solver	1	1.32	2.04	1.65	5.85
Fast sequence alignment	1	1.28	2.15	1.89	9.27
DNA sequence mapping	1	1.46	1.96	2.21	6.54
Neural network	1	1.88	3.21	2.67	11.85
Dijkstra	1	1.22	1.35	1.05	2.53
Epidemic simulation	1	1.09	1.77	2.04	8.34
Molecular dynamics	1	1.15	2.20	1.6	7.68
Graph partitioning	1	1.27	2.65	2.45	14.47
Euler equation solver	1	1.33	2.85	2.5	6.74
Evolutionary algorithm	1	1.54	2.54	2.2	6.2
IO proxy application	1	1.32	2.96	2.56	5.88
Mesh refinement application	1	1.65	2.33	2.75	6.32
CNN	1	1.13	1.91	2.24	7.69
Poisson equation solver	1	1.08	1.78	2.53	8.52
Monte Carlo kernel	1	1.24	2.12	2.64	6.03
HACC	1	1.35	2.52	2.31	7.84
Radiative transfer solver	1	1.42	2.22	1.75	8.93

2 Reviewer #2

We thank the reviewer for the suggestions. We have summarized the major modifications in the revised manuscript based on your comments and answered each comment as follows.

Major modifications:

- We have added the discussion of graph partitioning and further refinement based on modularity.
 - We have added the discussion of multifractal analysis on graphs.
1. **a. The problem of automatic parallelization/partitioning has been well-studied and is notoriously difficult for languages with arbitrary aliasing, like C/OpenCL. The authors don't discuss how they address mitigate issues with aliasing and ensure the correctness of the partition. b. That dataflow can be used to partition software is known since the 1970s and was well studied also in the 1980s. It is not clear, however, how the partitioning presented deals with issues of correctness due to the partition.**

In C, C++, and some other programming languages, the term aliasing refers to a situation where two different expressions or symbols refer to the same object. When references access that object in different ways — as both reads and stores — there are consequences for the order in which these mixed accesses can happen. We rely on the LLVM compiler to mitigate issues with aliasing [ali].

In dynamic execution graphs, the correctness is maintained at different levels: instruction level, partitioning level, and mapping level.

At the instruction level, we perform flow dependency analysis as shown in Figure 2 to maintain the correct dependency, meaning one instruction depends on another instruction in terms of the control flow, data flow, and call flow. The control flow analyzes the dependencies among instructions that involve either jump or branch to make sure the one instruction has to be executed before the next one. The data flow analyzes the dependencies among variables where source registers of one instruction depend on the destination register of the previous one. The call flow analyzes the function calls to make sure the caller function has to be reached before executing the callee function.

At the partitioning level, we maximize the following objective to minimize the communication overhead.

$$M = \frac{1}{m}R_1 - \frac{1}{e}R_2 - \frac{1}{m}R_3 \quad (1)$$

$$R_1 = \sum_{i,j} \left[\underbrace{A_{ij}\delta(c_i, c_j)}_{\text{the sum of weights in a cluster}} - \underbrace{\frac{k_i^{in}k_j^{out}}{m}\delta(c_i, c_j)}_{\text{inter-cluster weight sum}} \right] \quad (2)$$

$$R_2 = \sum_{i,j \in CG} \underbrace{1(A_{ij} \neq 0)}_{\text{an edge exists}} \underbrace{1(d_{DFS}(j) > d_{DFS}(i))}_{\text{the edge is backward}} \quad (3)$$

$$R_3 = \sum_{c=1}^{n_c} |A_c - A_{c'}| \quad (4)$$

where m is the sum of weights of all edges ($m = \sum_{i,j} w_{ij}$); e is the number of edges; A_{ij} represents the edge weight from node j to i in a graph, 0 means no edge exists; B_{ij} represents the edge weight from cluster j to i in a CG; k_i^{in} represents the sum of weights of all in-coming edges adjacent to node i ($k_i^{in} = \sum_p w_{pi}$); k_i^{out} represents the sum of weights of all out-going edges adjacent to node i ($k_i^{out} = \sum_q w_{iq}$); the delta function $\delta(u, v)$ equals 1 if $u = v$, and 0 otherwise; c_i is a cluster index from 1 to n ; $1(s)$ is the indicator function. It equals 1 if s evaluates true, and 0 otherwise; l_1 and l_2 are user-defined parameters to indicate the number of in-coming and out-going edges each cluster should have; $d_{DFS}(i)$ measures the depth of node i via depth first

search (DFS); n_c represents the number of clusters; A_c is the sum of weights of all edges within cluster c ($A_c = \sum_{i,j \in c} w_{ij}$); and c' is the cluster connected to cluster c .

Note that the second term is used to prevent cyclic dependencies in the clusters that would lead to deadlock when mapping on hardware. The idea behind this is whenever there is a direct edge from node i to node j ($A_{ij} \neq 0$), the depth of node j ($d_{DFS}(j)$) cannot be larger than the depth of node i ($d_{DFS}(i)$).

At the mapping level, we sort the clusters in a topological order that guarantees a cluster i is first scheduled ahead of cluster j if there is an edge from cluster i to cluster j .

2. **a. Given performance estimation for individual computational nodes, the mapping problem for data flow graphs is a well-studied research area. The authors don't acknowledge, discuss nor compare to established methods in this domain. b. The description is a mischaracterization of the state of research. Applications are distinct from computational kernels, and the mapping problem for (dataflow) applications is a well-established and studied method, as described in the general comments above. A useful, albeit dated survey can be found in "Mapping on multi/many-core systems: survey of current and emerging trends" by Kumar Singh, Shafique, Kumar and Henkel in DAC'13. Since then, several groups have continued to iterate on this problem and advanced the state of the art, e.g. the groups of J. Teich, J. Castrillon, A. Pimentel, M. Glass, among others.**

We have described the state-of-the-art mapping techniques for data flow graphs in the manuscript, which is reproduced as follows.

The scheduling and mapping of dataflow graphs are a well-studied research area including synchronous dataflow [lee87, LM87] and dynamic dataflow [buc93]. [SL93a, SL93b] extend the job-shop scheduling techniques to account for interprocessor communication costs. Pino et al. [PPL94] show how to construct schedules for heterogeneous multiprocessors. Falk et al. [FKH⁺08] give a parallel scheduling strategy based on clustering and demonstrate significant performance gains for multimedia applications.

Dependency Analysis

Figure 2: Flow dependency analysis.

Note that these dataflow graphs in the literature are coarse-grained as each node represents a function in a program and each edge represents a signal path. However, each node in a dynamic execution graph introduced in this manuscript represents one LLVM IR instruction. It is coarse-grained enough to reduce simulation time and memory space for keeping track of all low-level assembly instructions and data structures. At the same time, It is fine-grained enough to express inter-dependencies between each pair of instructions dynamically collected. Each edge represents data / control / call dependencies as shown in Figure 2. The control flow analyzes the dependencies among instructions that involve either jump or branch to make sure the one instruction has to be executed before the next one. The data flow analyzes the dependencies

among variables where the source registers of one instruction depend on the destination register of the previous one. The call flow analyzes the function calls to make sure the caller function has to be reached before executing the callee function. Therefore, the graph incorporates data-flow, control-flow, and call-flow that closely match the data structures used traditionally in inter-procedural data flow analysis.

We have compared our framework with the traditional mapping trends on multi-core systems [SSKH13] as follows.

- The applications in [SSKH13] are mostly related to dataflow such as signal processing whereas we don't have this assumption. The applications can be any programs that are written in a high-level language that is able to collect LLVM IR.
- The target platform in [SSKH13] is multi-core systems whereas we focus on the heterogeneous system that consists of GPUs and CPUs.
- The dynamic execution graphs used in this paper is much more fine-grained compared to the dataflow graphs where each node is a function or program. Each node in a dynamic execution graph is one LLVM IR instruction.
- Due to the different structures in graphs, we also consider the partitioning of graphs into different clusters that can be most beneficial to the heterogeneous system.

3. It is unclear whether, how much or how the method considers the communication overhead for mapping multiple kernels to a dataflow system.

We have added the discussion of the communication overhead in the result section, which is reproduced below.

Once multiple kernels are partitioned from the dynamic execution graph from a high-level application, we further refine the partitions to minimize the communication overhead [XXNB17, XNB21]. The partitioning problem can be formulated as an objective function [New06] to be maximized as follows.

$$M = \frac{1}{m}R_1 - \frac{1}{e}R_2 - \frac{1}{m}R_3 \quad (5)$$

$$R_1 = \sum_{i,j} \left[\underbrace{A_{ij}\delta(c_i, c_j)}_{\text{the sum of weights in a cluster}} - \underbrace{\frac{k_i^{in}k_j^{out}}{m}\delta(c_i, c_j)}_{\text{inter-cluster weight sum}} \right] \quad (6)$$

$$R_2 = \sum_{i,j \in CG} \underbrace{1(A_{ij} \neq 0)}_{\text{an edge exists}} \underbrace{1(d_{DFS}(j) > d_{DFS}(i))}_{\text{the edge is backward}} \quad (7)$$

$$R_3 = \sum_{c=1}^{n_c} |A_c - A_{c'}| \quad (8)$$

where m is the sum of weights of all edges ($m = \sum_{i,j} w_{ij}$); e is the number of edges; A_{ij} represents the edge weight from node j to i in an SPDG, 0 means no edge exists; B_{ij} represents the edge weight from cluster j to i in a CG; k_i^{in} represents the sum of weights of all in-coming edges adjacent to node i ($k_i^{in} = \sum_p w_{pi}$); k_i^{out} represents the sum of weights of all out-going edges adjacent to node i ($k_i^{out} = \sum_q w_{iq}$); the delta function $\delta(u, v)$ equals 1 if $u = v$, and 0 otherwise; c_i is a cluster index from 1 to n ; $1(s)$ is the indicator function. It equals 1 if s evaluates true, and 0 otherwise; l_1 and l_2 are user-defined parameters to indicate the number of in-coming and out-going edges each cluster should have; $d_{DFS}(i)$ measures the depth of node i via depth first search (DFS); n_c represents the number of clusters; A_c is the sum of weights of all edges within cluster c ($A_c = \sum_{i,j \in c} w_{ij}$); and c' is the cluster connected to cluster c .

In order to maximize the objective function M , the first term R_1 should be also be maximized whereas the second and third terms R_2 and R_3 should be minimized. The maximization of R_1 implies that we want to maximize the sum of weights in a cluster while minimizing the inter-cluster weight sum (data communication). The second term is used to prevent cyclic dependencies in the clusters that would lead to deadlock when mapping on hardware. The idea behind this is

whenever there is a direct edge from node i to node j ($A_{ij} \neq 0$), the depth of node j ($d_{DFS}(j)$) cannot be larger than the depth of node i ($d_{DFS}(i)$). The third term is used to balance the workloads between different clusters to prevent one cluster contains most of the instructions (A_c) while others in parallel only have a few instructions ($A_{c'}$).

We adopt the algorithm used in [BGLL08] that is repeated iteratively. First, each node is in its cluster according to GAE partitioning in the initial partition. Next, for each node i , we consider the neighbors i' of i and we evaluate the gain of the objective function ΔM that would take place by removing i from its community and by placing it in the community of i' , i.e. $\Delta M = M(P_{new}) - M(P_{old})$ where P_{old} represents node i in its original cluster and P_{new} represents node i in the community of i' . For each neighbor, we can calculate the gain. Then, node i is placed in the community of the neighbor that has the highest gain when the gain is positive. Otherwise, i stays in its original community. This process is applied repeatedly and iteratively for all nodes until no further gain improvement can be achieved.

4. **The authors correctly observe that dynamic information is central to the CPU/GPU classification problem from [9, 22, 23, 27], but do not compare to other performance estimation methods using dynamic information (e.g. would the model beat LLVM-MCA on these unrolled dynamic execution graph? what about other more sophisticated performance estimation methods based on profiling?).**

In LLVM-MCA, we collect some statistics, i.e., the number of iterations and total cycles as inputs to a neural network. However, compared to previous approaches, the accuracy is only 48.6%. We believe it is because of too few useful statistics from LLVM-MCA. We also look into some more sophisticated performance estimation methods based on profiling such as [KC17, gpu], but it cannot be used in our datasets.

5. **What is "the" universal IR?**

LLVM is a language-independent type system that exposes the primitives used to implement high-level language (HLL) features. It includes an instruction for typed address arithmetic and a mechanism for implementing the exception handling HLL features [LA04]. Moreover, LLVM IR is an abstract machine language that performs the basic computations, memory operations, and branch instructions with unlimited virtual registers to prevent register spilling. Compared to assembly, the advantage of IR is that it is in the static single assignment (SSA) form. It means a variable can be defined only once and its value can never change, which simplifies compiler optimizations to a very significant degree. Therefore, we consider it universal to represent any high-level program. Nevertheless, we make it clearer by rephrasing it to "low-level virtual machine intermediate representation" in the abstract.

6. **Are algorithms really becoming more complex? Perhaps the requirements, or the size of systems, but the algorithms themselves I'm not sure.**

Yes, we think some algorithms are becoming more complex, not in terms of computational complexity, but in terms of structural complexity. Take matrix multiplication as an example. The standard method to implement matrix multiplication is to have 3 for loops to iterate over each element in a matrix to perform multiplication, leading to $O(N^3)$ where N is the size of a matrix. The Strassen algorithm is usually faster than the standard matrix multiplication algorithm for large matrices, with a better asymptotic complexity, leading to $O(N^{2.8})$ time complexity. It is a recursive method where the matrix is divided into 4 sub-matrices of dimensions $N/2 \times N/2$ in each recursive step. The most recent implementation is to use a deep reinforcement learning approach based on AlphaZero for discovering efficient and provably correct algorithms for the multiplication of arbitrary matrices [FBH⁺22, alp]. It can discover algorithms that outperform the state-of-the-art complexity for many matrix sizes. The agent, AlphaTensor, is trained to play a single-player game where the objective is to find tensor decompositions within a finite factor space. Figure 3 shows the evolution of these algorithms to show the structural complexity.

7. **Why such an emphasis on load balancing? Load balancing plays a role in achieving high performance or energy efficiency, as do many other things: good mappings, ILP, compilation, algorithmic improvements/overheads, efficient use of accelerators,**

Figure 3: Evolution of the structural complexity of matrix multiplication.

good cache behavior/locality, etc, all arguably play at least as big a role as load balancing, if not bigger.

We have adapted the introduction section to include many other things in the discussion (good mappings, ILP, compilation, algorithmic improvements/overheads, efficient use of accelerators, good cache behavior/locality, etc), which is reproduced below.

To manage the need for computational gains, heterogeneous systems require intelligent, flexible, and efficient programming strategies that can match the requirements of real-world applications to the strengths of the heterogeneous architecture. This matching should achieve better good mappings, compiler transformations [BGS94], efficient use of accelerators [BBD⁺12], good cache locality [AJS⁺17], and load balancing [XXNB17] to provide high performance and energy efficiency [MV15, XNB19]. However, the existing monolithic programming models and task mapping to compute platforms do not fully exploit the recent heterogeneity as well as architectural innovations in current hardware systems. It cannot efficiently use the heterogeneous processing elements and could exacerbate the load imbalance and communication inefficiencies [MV15, EBA⁺11, XXNB17]. For example, the conventional CPU-only or GPU-only optimization techniques may not be suitable for a heterogeneous system that combines both of them. This is due to the architectural and programming model differences of these hardware accelerators. Therefore, novel optimization approaches are required to realize the potential of heterogeneous systems and achieve the goals of exascale performance.

8. **Figure 1: Are frequent system and firmware updates really the main problem? I would strongly contend that. Whether these things make our lives better is debatable, and this debate is certainly outside the scope of this paper.**

We have rephrased the caption of Figure 1 and made a few changes in Figure 1 to address the reviewer’s concern. Below is Figure 4 as also shown in the manuscript.

9. **”Formal” has a well-defined meaning in computer science (using a formal language/methods); the definition of the device mapping problem you give/cite there is not formal.**

We have removed the word ”formal” or ”formally”.

10. **The authors discuss recent developments in code representation for machine learning without making clear that this refers specifically to this domain.**

Recently, various graph representations were proposed for machine learning to represent and capture the latent information flow in a program (e.g., abstract syntax tree (AST) [AZLY19], contextual flow graph (XFG) [BNJH18b], and control and data flow graph (CDFG) [BGEC20]).

Figure 4: Autonomous heterogeneous computing system. The recent advance of technologies enables the fast progress of autonomous cars and unmanned aerial vehicles(a). However, with the commonly used system components such as the controller and convolutional neural networks for image recognition (b), parallelization and communication overhead become inevitable concerns for programmers as the complicated and ever-changing software needs to be parallelized and executed on a heterogeneous system (c). The proposed framework makes the manual process autonomous without human intervention by profiling applications (d), constructing dynamic dataflow graphs (e), and mapping kernels onto the platform via machine learning models (f).

11. While the plural of code, codes, is perfectly grammatical, it is not usually used to refer to multiple pieces of software code. Codes gives the impression of there being different encodings, like when people talk about different error-correcting codes, where the code (i.e. encoding) itself changes, and not the content of the coded artifact.

We have changed "codes" to "code" in the manuscript.

12. Dynamic dataflow graphs have an established meaning that is used in a whole field of research. For a standard reference discussing dynamic dataflow alongside other dataflow models, see for example "Ptolemaeus, Claudius, ed. System design, modeling, and simulation: using Ptolemy II. Vol. 1. Berkeley: Ptolemy. org, 2014."

In order to distinguish our graphs from dynamic dataflow graphs in the literature, we change it to dynamic execution graphs.

13. Figure 3: the graphs a) b) and c) are nice visualizations; however, it is unclear what insight we can gain from such a visualization at this point. If it is the fractal properties, this is not well explained here, nor is the algorithm used to visualize them. Does the fractal property depend on this visualization algorithm? A closer (zoomed in) view showing the actual labels for nodes and edges of the graph would be much more insightful, as it would give an idea of the actual structure of the graph.

We have added some common zoomed-in graph patterns among dynamic execution graphs with high-level C code in Figure 3 as shown in Figure 5. Loops are commonly used in any programming language that can execute a group of statements multiple times. When arrays are used inside a loop statement, the corresponding dynamic execution graph has a star shape. The central node that is connected to different branches is the "getelementptr" LLVM IR. It is used to get the address of a sub-element of an aggregate data structure. Each branch corresponds to different instances of $a[i] = i$. When none of the arrays are used inside a loop, the corresponding dynamic

Figure 5: Dynamic execution graphs and multifractal properties. Figure (a), (b), and (c) shows basic graph patterns in graphs where the code contains either loops or sequential statements. Figure (d), (e), and (f) shows the constructed code graphs for sequence alignment, signal processing, and convolutional neural network, respectively. The graphs are a hybrid of fundamental graph patterns in (a-c). Figure (g) shows the multifractal spectrum and some definitions such as α_0 and spectrum width w . Figure (h) shows how a generalized fractal dimension for a graph looks like. Figure (i) shows three multifractal spectra for (d-f) to demonstrate multifractal spectrum can identify the heterogeneous graph structures in different dynamic execution graphs.

execution graph has a mesh shape. When only sequential statements such as if-else are used in code, the corresponding dynamic execution graph has a tree shape to represent the information flow from the beginning to the end.

There are some insights we can gain from (d), (e), and (f).

- Each graph contains multiple fundamental graph patterns in (a), (b), and (c). For example, (d) clearly shows the mesh topology (b) and (e) has a star-shaped subgraph (a) that indicates the use of loops with arrays.
- In order to quantify the structural difference among the graphs, we analyze the multifractal spectra of the graphs in (i), which validates that multifractal analysis is able to detect the topological structures in graphs. This helps us to design the feature extraction algorithm based on multifractal analysis in PGL.

We saved the graphs in the gexf format and visualize them in Gephi using ForceAtlas2 layout [JVHB14]. The fractal properties do not depend on the visualization algorithm. Below you can find a brief discussion on the multifractal properties such as spectrum and generalized fractal dimension.

Real-world fractals may not be homogeneous, meaning that there is rarely an identical motif repeated on all scales. Therefore, multifractal analysis is developed to investigate self-repeating patterns at different scales of complex networks. In multifractal analysis, the fixed-size box-counting algorithm is commonly used to find the multifractal properties [HJK⁺86]. For a given probability measure $0 \leq \mu \leq 1$, we consider the partition

$$Z_r(q) = \sum_{\mu(B) \neq 0} [\mu(B)]^q \quad (9)$$

where q is a distortion exponent and the sum runs over all different non-overlapping boxes B of a given box size r . The mass exponent function $\tau(q)$ of the measure μ is defined as

$$\tau(q) = \lim_{r \rightarrow 0} \frac{\ln Z_r(q)}{\ln r} \quad (10)$$

The generalized fractal dimension of the measure μ is defined as

$$D_q = \frac{\tau(q)}{q-1}, q \neq 1 \quad (11)$$

The singularity spectrum $f(\alpha)$ with the Holder exponent α and the mass exponent function $\tau(q)$ are connected via the Legendre transform [MM82].

$$\alpha(q) = \frac{d\tau(q)}{dq} \quad (12)$$

$$f(\alpha) = q\alpha(q) - \tau(q) \quad (13)$$

For a network, the measure μ of each box B is defined as the ratio of the number of nodes covered by the box ($N(B)$) and the total number of nodes in the network N .

$$\mu(B) = \frac{N(B)}{N} \quad (14)$$

14. **a. Figure 4: is this really necessary? I did not find the graph very insightful, it shows some linear regressions for some particular parameters on the model for some examples. More generally, Multifractal analysis is not a well-established method that the authors can expect a reader to know of, nor are Lipschitz-Holder exponents, etc. While the authors do explain some of these concepts, they do not really explain what the point is of showing these details of the modeling. It seems to be an**

unnecessary level of detail, given other omissions in the paper (like the runtime system for the partitioned graphs, the discussion and comparison of the mapping literature, etc.) b. The analysis of the 132 programs and description of their fractal properties seems to be interesting, but again, I did not understand the point of it (see above). What insight does this give me into end-to-end compilation in general, the overall goal of the paper?

Below you can find the discussion of multifractal analysis on graphs.

Real-world fractals may not be homogeneous, meaning that there is rarely an identical motif repeated on all scales. Therefore, multifractal analysis is developed to investigate self-repeating patterns at different scales of complex networks. In multifractal analysis, the fixed-size box-counting algorithm is commonly used to find the multifractal properties [HJK⁺86]. For a given probability measure $0 \leq \mu \leq 1$, we consider the partition

$$Z_r(q) = \sum_{\mu(B) \neq 0} [\mu(B)]^q \quad (15)$$

where q is a distortion exponent and the sum runs over all different non-overlapping boxes B of a given box size r . The mass exponent function $\tau(q)$ of the measure μ is defined as

$$\tau(q) = \lim_{r \rightarrow 0} \frac{\ln Z_r(q)}{\ln r} \quad (16)$$

The generalized fractal dimension of the measure μ is defined as

$$D_q = \frac{\tau(q)}{q-1}, q \neq 1 \quad (17)$$

The singularity spectrum $f(\alpha)$ with the Holder exponent α and the mass exponent function $\tau(q)$ are connected via the Legendre transform [MM82].

$$\alpha(q) = \frac{d\tau(q)}{dq} \quad (18)$$

$$f(\alpha) = q\alpha(q) - \tau(q) \quad (19)$$

For a network, the measure μ of each box B is defined as the ratio of the number of nodes covered by the box ($N(B)$) and the total number of nodes in the network N .

$$\mu(B) = \frac{N(B)}{N} \quad (20)$$

There are a few insights we can learn from this figure.

- It provides a universal model that builds the relationship between the graph properties such as the multifractal spectrum and the system-level metrics such as the parallelization degree and communication overhead. If such a model can accurately capture the relationship, the optimal degree of parallelization can be calculated by the graph properties, without the manual tuning from a programmer.
 - Similar to Figure 3(i), it provides us with what design choices we can gain to develop the feature extraction algorithm. PGL contains a feature extraction algorithm used by the graph neural network to predict a label. The figure shows that multifractal analysis can capture the graph topological structures, which can be further used in the feature extraction algorithm.
15. **The language in the methods section is clearly distinct from the language before, showing that less care was taken for the proof reading (e.g. several articles are missing before nouns, etc)**

We proofread the entire manuscript and fixed these issues.

Figure 6: The impact of training on the performance of different applications.

16. **Figure 6:** this three-dimensional plot is hard to read. In general it is hard to see three-dimensional visualizations in a two-dimensional sheet of paper. I would recommend showing the two projections individually instead.

We have modified the 3D plot into 2 2D plots as shown in Figure 6.

17. **The comparison to state of the art is confusing:** how is gem5-aladdin, a simulator, a comparison to PGL? More importantly, how is this, and a generic neural network with reinforcement learning state-of-the-art? It certainly is not state of the art in mapping (see above), which explains how the numbers of the speedup can be so high!

The traditional mapping of graphs is not applicable mainly because of the difference between graphs. The dataflow graphs in the literature are much coarser-grained compared to the dynamic execution graphs. In dataflow graphs, each node is either a function block or a kernel that consists of a sequence of instructions to be run on hardware. In dynamic execution graphs, each node is one LLVM IR instruction. Therefore, the mapping introduced in this manuscript has two folds. First, each graph has to be partitioned into clusters / kernels that represent a sequence of instructions. Second, each kernel is mapped onto a heterogeneous platform.

Gem5-aladdin provides end-to-end SoC simulation by integrating the gem5 system simulator with the Aladdin accelerator simulator. It provides a framework to accelerate applications. Therefore, we choose gem5-aladdin as a comparison to our work.

References

- [AJS⁺17] Shaizeen Aga, Supreet Jeloka, Arun Subramaniyan, Satish Narayanasamy, David Blaauw, and Reetuparna Das. Compute caches. In *2017 IEEE International Symposium on High Performance Computer Architecture (HPCA)*, pages 481–492. IEEE, 2017.
- [ali] LLVM Alias Analysis Infrastructure. <https://llvm.org/docs/AliasAnalysis.html>.
- [alp] Alphasensor Software. <https://github.com/deepmind/alphasensor>.
- [amd] Algebraic multigrid benchmark. <https://github.com/LLNL/AMG>.
- [amr] MiniAMR Adaptive Mesh Refinement (AMR) Mini-App. <https://github.com/Mantevo/miniAMR>.
- [AZLY19] Uri Alon, Meital Zilberstein, Omer Levy, and Eran Yahav. code2vec: Learning distributed representations of code. *Proceedings of the ACM on Programming Languages*, 3(POPL):1–29, 2019.
- [BBD⁺12] George Bosilca, Aurelien Bouteiller, Anthony Danalis, Thomas Herault, Pierre Lemariniere, and Jack Dongarra. Dague: A generic distributed dag engine for high performance computing. *Parallel Computing*, 38(1-2):37–51, 2012.

- [BGEC20] Alexander Brauckmann, Andrés Goens, Sebastian Ertel, and Jeronimo Castrillon. Compiler-based graph representations for deep learning models of code. In *Proceedings of the 29th International Conference on Compiler Construction*, pages 201–211, 2020.
- [BGLL08] Vincent D Blondel, Jean-Loup Guillaume, Renaud Lambiotte, and Etienne Lefebvre. Fast unfolding of communities in large networks. *Journal of statistical mechanics: theory and experiment*, 2008(10):P10008, 2008.
- [BGS94] David F Bacon, Susan L Graham, and Oliver J Sharp. Compiler transformations for high-performance computing. *ACM Computing Surveys (CSUR)*, 26(4):345–420, 1994.
- [BNJH18a] Tal Ben-Nun, Alice Shoshana Jakobovits, and Torsten Hoefler. Neural code comprehension: A learnable representation of code semantics. In S. Bengio, H. Wallach, H. Larochelle, K. Grauman, N. Cesa-Bianchi, and R. Garnett, editors, *Advances in Neural Information Processing Systems 31*, pages 3588–3600. Curran Associates, Inc., 2018.
- [BNJH18b] Tal Ben-Nun, Alice Shoshana Jakobovits, and Torsten Hoefler. Neural code comprehension: A learnable representation of code semantics. *arXiv preprint arXiv:1806.07336*, 2018.
- [bow] Fast and sensitive gapped read aligner. <https://github.com/BenLangmead/bowtie2>.
- [buc93] Scheduling dynamic dataflow graphs with bounded memory using the token flow model. In *1993 IEEE international conference on acoustics, speech, and signal processing*, volume 1, pages 429–432. IEEE, 1993.
- [bwa] Burrow-Wheeler Aligner for short-read alignment. <https://github.com/lh3/bwa>.
- [CFBN⁺21] Chris Cummins, Zacharias Fisches, Tal Ben-Nun, Torsten Hoefler, Michael O’Boyle, and Hugh Leather. ProGraML: A Graph-based Program Representation for Data Flow Analysis and Compiler Optimizations. In *International Conference on Machine Learning (ICML)*, 2021.
- [cnn] CNN in face detection. <https://github.com/ShiqiYu/libfacedetection>.
- [CPWL17] Chris Cummins, Pavlos Petoumenos, Zheng Wang, and Hugh Leather. End-to-end deep learning of optimization heuristics. In *2017 26th International Conference on Parallel Architectures and Compilation Techniques (PACT)*, pages 219–232. IEEE, 2017.
- [cum] DeepTune software. <https://github.com/ChrisCummins/paper-end2end-dl>.
- [DA97] Ali Dasdan and Cevdet Aykanat. Two novel multiway circuit partitioning algorithms using relaxed locking. *IEEE Transactions on Computer-Aided Design of Integrated Circuits and Systems*, 16(2):169–178, 1997.
- [dee] DeepLLVM software. <https://gitlab.com/ecs-lab/deepllvm>.
- [dij] Dijkstra shortest path. <https://github.com/Lehmannhen/MPI-Dijkstra>.
- [EBA⁺11] Hadi Esmaeilzadeh, Emily Blem, Renee St Amant, Karthikeyan Sankaralingam, and Doug Burger. Dark silicon and the end of multicore scaling. In *2011 38th Annual international symposium on computer architecture (ISCA)*, pages 365–376. IEEE, 2011.
- [ees] High-order Lagrangian Hydrodynamics Miniapp. <https://github.com/CEED/Laghos>.
- [epi] Epidemic simulation software. https://github.com/ZiluTian/epidemic_simulation.
- [evo] Lamarckian evolutionary algorithm for de novo drug design. <https://github.com/UAMCAntwerpen/LEADD>.
- [FBH⁺22] Alhussein Fawzi, Matej Balog, Aja Huang, Thomas Hubert, Bernardino Romera-Paredes, Mohammadamin Barekatin, Alexander Novikov, Francisco J R Ruiz, Julian Schrittwieser, Grzegorz Swirszcz, et al. Discovering faster matrix multiplication algorithms with reinforcement learning. *Nature*, 610(7930):47–53, 2022.

- [FKH⁺08] Joachim Falk, Joachim Keinert, Christian Haubelt, Jürgen Teich, and Shuvra S Bhattacharyya. A generalized static data flow clustering algorithm for mp soc scheduling of multimedia applications. In *Proceedings of the 8th ACM international conference on Embedded software*, pages 189–198, 2008.
- [FLNG⁺20] Neil M Ferguson, Daniel Laydon, Gemma Nedjati-Gilani, Natsuko Imai, Kylie Ainslie, Marc Baguelin, Sangeeta Bhatia, Adhiratha Boonyasiri, Zulma Cucunubá, Gina Cuomo-Dannenburg, et al. Impact of non-pharmaceutical interventions (npis) to reduce covid-19 mortality and healthcare demand. 2020.
- [FYP⁺21] Yuantao Feng, Shiqi Yu, Hanyang Peng, Yan-Ran Li, and Jianguo Zhang. Detect faces efficiently: A survey and evaluations. *IEEE Transactions on Biometrics, Behavior, and Identity Science*, 4(1):1–18, 2021.
- [GML⁺16] Jing Gong, Stefano Markidis, Erwin Laure, Matthew Otten, Paul Fischer, and Misun Min. Nekbone performance on gpus with openacc and cuda fortran implementations. *The Journal of Supercomputing*, 72(11):4160–4180, 2016.
- [gpu] A GPU performance prediction toolkit for CUDA programs. <https://github.com/ekondis/gpuroofperf-toolkit>.
- [hac] Stand-alone version of HACC’s distributed-memory, pencil-decomposed, parallel 3D FFT. <https://xgitlab.cels.anl.gov/hacc/SWFFT>.
- [HJK⁺86] Thomas C Halsey, Mogens H Jensen, Leo P Kadanoff, Itamar Procaccia, and Boris I Shraiman. Fractal measures and their singularities: The characterization of strange sets. *Physical review A*, 33(2):1141, 1986.
- [ime] IMEX transport mini-app. https://github.com/ECP-Astro/thornado_mini.
- [iop] Multi-purpose, Application-Centric, Scalable I/O Proxy Application. <https://github.com/LLNL/MACSiO>.
- [JVHB14] Mathieu Jacomy, Tommaso Venturini, Sebastien Heymann, and Mathieu Bastian. Forceatlas2, a continuous graph layout algorithm for handy network visualization designed for the gephi software. *PloS one*, 9(6):e98679, 2014.
- [KC17] Elias Konstantinidis and Yiannis Cotronis. A quantitative roofline model for gpu kernel performance estimation using micro-benchmarks and hardware metric profiling. *Journal of Parallel and Distributed Computing*, 107:37–56, 2017.
- [KDW22] Alan Kerstjens and Hans De Winter. Leadd: Lamarckian evolutionary algorithm for de novo drug design. *Journal of cheminformatics*, 14(1):1–20, 2022.
- [KW16] Thomas N Kipf and Max Welling. Semi-supervised classification with graph convolutional networks. *arXiv preprint arXiv:1609.02907*, 2016.
- [LA04] Chris Lattner and Vikram Adve. Llvm: A compilation framework for lifelong program analysis & transformation. In *International symposium on code generation and optimization, 2004. CGO 2004.*, pages 75–86. IEEE, 2004.
- [lee87] Synchronous data flow. *Proceedings of the IEEE*, 75(9):1235–1245, 1987.
- [Li13] Heng Li. Aligning sequence reads, clone sequences and assembly contigs with bwa-mem. *arXiv preprint arXiv:1303.3997*, 2013.
- [LM87] Edward Ashford Lee and David G Messerschmitt. Static scheduling of synchronous data flow programs for digital signal processing. *IEEE Transactions on computers*, 100(1):24–35, 1987.
- [LPS⁺15] Dominique LaSalle, Md Mostofa Ali Patwary, Nadathur Satish, Narayanan Sundaram, Pradeep Dubey, and George Karypis. Improving graph partitioning for modern graphs and architectures. In *Proceedings of the 5th Workshop on Irregular Applications: Architectures and Algorithms*, pages 1–4, 2015.

- [LTBZ15] Yujia Li, Daniel Tarlow, Marc Brockschmidt, and Richard Zemel. Gated graph sequence neural networks. *arXiv preprint arXiv:1511.05493*, 2015.
- [LTPS09] Ben Langmead, Cole Trapnell, Mihai Pop, and Steven L Salzberg. Ultrafast and memory-efficient alignment of short dna sequences to the human genome. *Genome biology*, 10(3):1–10, 2009.
- [md] Molecular dynamics proxy application. <https://github.com/ECP-copa/ExaMiniMD>.
- [MM82] Benoit B Mandelbrot and Benoit B Mandelbrot. *The fractal geometry of nature*, volume 1. WH freeman New York, 1982.
- [MV15] Sparsh Mittal and Jeffrey S Vetter. A survey of cpu-gpu heterogeneous computing techniques. *ACM Computing Surveys (CSUR)*, 47(4):1–35, 2015.
- [ncc] NCC software. <https://github.com/spcl/ncc>.
- [nek] Nekbone software. <https://github.com/AMDComputeLibraries/Nekbone>.
- [New06] Mark EJ Newman. Modularity and community structure in networks. *Proceedings of the national academy of sciences*, 103(23):8577–8582, 2006.
- [nn] Open source neural network. <https://github.com/QuentinAM/neural-network-c>.
- [NNPN18] Anh Tuan Nguyen, Trong Duc Nguyen, Hung Dang Phan, and Tien N Nguyen. A deep neural network language model with contexts for source code. In *2018 IEEE 25th International Conference on Software Analysis, Evolution and Reengineering (SANER)*, pages 323–334. IEEE, 2018.
- [par] Multi-way graph partitioning algorithms. <https://github.com/alidasdan/graph-partitioning-algorithms>.
- [PBBA21] Emanuele Parisi, Francesco Barchi, Andrea Bartolini, and Andrea Acquaviva. Making the most of scarce input data in deep learning-based source code classification for heterogeneous device mapping. *IEEE Transactions on Computer-Aided Design of Integrated Circuits and Systems*, 41(6):1636–1648, 2021.
- [pgc] ProGraML software. <https://github.com/ChrisCummins/ProGraML>.
- [PPL94] Jose Luis Pino, Thomas M Parks, and Edward A Lee. Automatic code generation for heterogeneous multiprocessors. In *Proceedings of ICASSP'94. IEEE International Conference on Acoustics, Speech and Signal Processing*, volume 2, pages II–445. IEEE, 1994.
- [SL93a] Gilbert C Sih and Edward A Lee. A compile-time scheduling heuristic for interconnection-constrained heterogeneous processor architectures. *IEEE transactions on Parallel and Distributed systems*, 4(2):175–187, 1993.
- [SL93b] Gilbert C Sih and Edward A Lee. Declustering: A new multiprocessor scheduling technique. *IEEE Transactions on Parallel and Distributed Systems*, 4(6):625–637, 1993.
- [SSKH13] Amit Kumar Singh, Muhammad Shafique, Akash Kumar, and Jörg Henkel. Mapping on multi/many-core systems: survey of current and emerging trends. In *Proceedings of the 50th Annual Design Automation Conference*, pages 1–10, 2013.
- [VCC⁺17] Petar Veličković, Guillem Cucurull, Arantxa Casanova, Adriana Romero, Pietro Lio, and Yoshua Bengio. Graph attention networks. *arXiv preprint arXiv:1710.10903*, 2017.
- [XNB19] Yao Xiao, Shahin Nazarian, and Paul Bogdan. Self-optimizing and self-programming computing systems: A combined compiler, complex networks, and machine learning approach. *IEEE Transactions on Very Large Scale Integration (VLSI) Systems*, 27(6):1416–1427, 2019.

- [XNB21] Yao Xiao, Shahin Nazarian, and Paul Bogdan. Plasticity-on-chip design: Exploiting self-similarity for data communications. *IEEE Transactions on Computers*, 70(6):950–962, 2021.
- [xsb] XSBench: The Monte Carlo Macroscopic Cross Section Lookup Benchmark. <https://github.com/ANL-CESAR/XSBench>.
- [XXNB17] Yao Xiao, Yuankun Xue, Shahin Nazarian, and Paul Bogdan. A load balancing inspired optimization framework for exascale multicore systems: A complex networks approach. In *2017 IEEE/ACM International Conference on Computer-Aided Design (ICCAD)*, pages 217–224. IEEE, 2017.

Reviewers' comments:

Reviewer #1 (Remarks to the Author):

The authors answer positively to all my previous comments.

Reviewer #2 (Remarks to the Author):

I thank the authors for their revisions and careful explanations of the changes and my questions and concerns in the first iteration. The manuscript has improved and is going in the right direction. I have two main concerns remaining: the comparison to state of the art (mapping/partitioning), see points 2, 4, and 17 and the discussion of the communication overhead (see points 2 and 3). If these two points are addressed, I'd be happy to accept the manuscript. I'll mark it as major revisions, however, because these are not points that I believe can be argued away, but rather require an additional evaluation to be done. In the following, I respond to the individual points from the rebuttal:

1. The authors did not answer my questions here nor in the manuscript. The question was about correctness.

Instead, the authors explain what the concepts are (if I'm asking about aliasing, why do you explain to me what aliasing is?) or describe some formulation as an optimization problem they used to partition. None of this addresses correctness.

LLVM has heuristics to find aliasing, but they're not guaranteed to be correct. From the discussion (or lack thereof), I presume it means your flow is conservative and will not partition when the analysis cannot determine whether a variable might point to a different one.

In either case, I would expect a discussion of the limitations of the approach and examples of code it won't be able to partition.

2. It's great that you acknowledge some work in this area. The second-newest paper you cite there is 29 years old, I'd hardly call that "state of the art".

In particular, the point of this comment is not to check a box nor have you cite a specific paper, it's to understand how your approach builds on and or extends existing work and frame it in that context.

You say that "The target platform in [SSKH13] is multi-core systems whereas we focus on the heterogeneous system that consists of GPUs and CPUs".

The former subsumes the latter: there's a significant amount of work of mapping to heterogeneous multi-cores, with DSPs, GPUs, and other accelerators.

It's true that there's a difference between on-chip latencies and off-chip heterogeneous systems that you mention, but those boundaries are blurring (see e.g. NVIDIA's newly announced "grace hopper" architecture)

To be frank, I don't particularly like the SSKH13 paper, it was just a reference that is somewhat comprehensive, there's been much work since.

The point you make about the granularity is a very valid one and should probably be in the paper (more

so than, say, enumerating some selection of 30-year-old papers).

If the different granularity changes the methods significantly, maybe you can explain how and how your methods can scale much better (which you'll need for much larger graphs like these).

It's interesting that you want to use a finer granularity for systems with a higher communication overhead; how can you get away with this, compared to the state of the art?

3. Thanks for addressing the communication overhead. I don't think describing your objective function and precise formulation is very useful.

It seems like an implementation detail, and I did not gain much new insight from the formulation.

While I applaud the completeness of defining terms you use like the Kronecker delta (which is very standard), this level of detail might obfuscate the main ideas.

More importantly, this discussion tells me a lot of details on how you tried to minimize the communication overhead, but not how much it actually was. My question was not how do you model it, it was how big is it, what percentage of the execution time does it represent? I would recommend removing this discussion on how you model it, and instead, measuring it and discussing how large the overhead was.

4. I frankly did not understand this response. Why as input to a neural network? Why can they not be used in your dataset? More concretely: A part of your system estimates the performance of a kernel by using dynamic information. My question is "how accurate is this?", compared to e.g. LLVM-MCA or other more sophisticated approaches. In other words, if you split up only your performance estimation, how much better does it estimate the performance of a piece of code? I'm not asking you to replace that part of your system with LLVM-MCA, I want to understand how good that part of the system is, compared to a standard baseline.

5. Thanks for the clarification. I know what LLVM and SSA are, and I think calling it a universal IR is controversial at best; this change clarifies and addresses my doubt nevertheless, it was not clear to mean that you meant LLVM by that.

6. Fair enough. The example is only somewhat convincing, as the Strassen algorithm is over 50 years old and was the state of the art until last year's AlphaZero result. Nevertheless, I get the spirit of the answer and I'm happy to accept this point.

7.-12. Great!

13. This is a significant improvement and insightful, thanks.

14. The explanation seems mathematically reasonable, although I'm not familiar with this topic. I think it's still not very insightful though. Computer engineers will not gain much from the mathematics there, whereas a geometer who studies fractals will not understand much about the compiler side. What's your target audience? I think if you're describing the application, you can focus on how the concepts are useful and intuitively explain what's happening, while safely referring to literature for the formal definitions and technical details. The goal of this is not to criticize your work, but rather, to suggest ways in which it can be made more accessible to your target audience, as I understand it.

15.-16. Great!

17. Why is the difference in graphs significant? That they were intended for a different granularity does not mean you cannot use the algorithm (maybe it will fare poorly, in which case that's a great motivation for your algorithm!) I cannot see how the granularity is (mathematically) relevant to the applicability of the algorithm. Moreover, as you explain, the second phase of your second "fold", mapping kernels to the heterogeneous platform very much sounds like the same granularity, doesn't it?

The point about the simulator was still not addressed, however, or I did not understand your explanation. On the one hand, you have gem5-aladdin which is a simulator and allows you to simulate an execution of an application running on a CPU/GPU heterogeneous system. On the other hand, you have PGL which does partitioning and mapping and then provides a result. I could see how you'd use gem5-aladdin to estimate the performance of a design point found by PGL, but gem5-aladdin does not do mapping and partitioning; it does not explore the design space and only evaluates a single point in it. How can this be a meaningful comparison? It's comparing oranges to apples, a simulator that evaluates one design point, versus an algorithm that chooses a design point.

Response to Referees

June 6, 2023

We thank all the reviewers for their feedback and suggestions for improvement. We followed closely the reviewers' feedback and include below our point-by-point responses. The comments from each reviewer are highlighted in bold, our responses to the reviewers' comments are discussed after each question/comment, and the modifications we made in the revised manuscript are highlighted in red.

1 Reviewer #2

We thank the reviewer for the constructive suggestions. We have summarized the major modifications in the revised manuscript based on your comments and answered each comment as follows. We are looking forward to hearing your feedback.

Major modifications:

- We performed experiments concerning the measurement of communication and computation times for each application in different frameworks.
- We performed experiments on different graph representations: dynamic data dependence graphs in Aladdin [SRWB14], control-flow/data-flow/call-flow graph, and dynamic execution graph.
- We added the discussion of the state-of-the-art approaches and rewrote the multifractal analysis to our target audience.

Detailed Comments

- [1] **The authors did not answer my questions here nor in the manuscript. The question was about correctness. Instead, the authors explain what the concepts are (if I'm asking about aliasing, why do you explain to me what aliasing is?) or describe some formulation as an optimization problem they used to partition. None of this addresses correctness. LLVM has heuristics to find aliasing, but they're not guaranteed to be correct. From the discussion (or lack thereof), I presume it means your flow is conservative and will not partition when the analysis cannot determine whether a variable might point to a different one. In either case, I would expect a discussion of the limitations of the approach and examples of code it won't be able to partition.**

Thank you very much for your comment and suggestion. Indeed, LLVM alias analysis cannot guarantee the correctness of memory dependencies. In fact, the LLVM aliasing analysis report shown in Figure 1 illustrates 3 types of aliases: MustAlias, MayAlias, and NoAlias. In the construction of the dynamic execution graphs, we add an edge if two memory objects are reported in MustAlias and MayAlias. If a MustAlias is encountered, then this indicates a new true memory dependency that should be added to the dynamic execution graph. If a MayAlias is encountered, then this indicates that the two memory objects could be dependent or not. In our implementation, we treat this MayAlias occurrence as MustAlias and insert an edge from node i to j in the dynamic execution graph. Please note that this is not harmful as node i sends unuseful data to node j as node j can just ignore it. It means that the tool is conservative in that it treats all of the guaranteed and possible memory dependencies as true dependencies. This could only lead to more edges in the graph.

```

MustAlias:   i32* %2, i8* %"Cast as void7"
MayAlias:   i32* %orig, i32* %sol
MayAlias:   i32* %filter, i32* %orig
MayAlias:   i32* %filter, i32* %sol
MayAlias:   i32* %orig, i8* %bufPos
MayAlias:   i32* %sol, i8* %bufPos
MayAlias:   i32* %filter, i8* %bufPos
MayAlias:   i32* %orig, i8* %storeBlock0
MayAlias:   i32* %sol, i8* %storeBlock0
MayAlias:   i32* %filter, i8* %storeBlock0
MayAlias:   i8* %bufPos, i8* %storeBlock0
MayAlias:   i32* %2, i32* %orig
MayAlias:   i32* %2, i32* %sol
NoAlias:    i32* %2, i32* %filter
MayAlias:   i32* %2, i8* %bufPos
MayAlias:   i32* %2, i8* %storeBlock0
MayAlias:   i32* %3, i32* %orig
MayAlias:   i32* %3, i32* %sol
NoAlias:    i32* %3, i32* %filter
MayAlias:   i32* %3, i8* %bufPos
MayAlias:   i32* %3, i8* %storeBlock0
NoAlias:    i32* %2, i32* %3

```

Figure 1: LLVM alias analysis report using flags "-basicaa -aa-eval -print-allalias-modref-info".

At the partitioning level, we agree with you that we cannot guarantee a correct partition of a graph, meaning a node which should have been placed into cluster i is placed in cluster j instead. However, the correctness of the graph partitioning doesn't lead to wrong results upon execution. The wrong results are usually caused by (1) missing instructions; (2) wrong order of instructions being executed; or (3) wrong data being fetched. However, none of the scenarios could happen in our setup of the dynamic execution graph because of the following safeguards that are placed: (1) Each graph contains all of the instructions and their direction dependencies, and the proposed partitioning does not remove the instructions and their dependencies. (2) The order of executing the instructions is preserved by exploiting our proposed topological sort strategy that guarantees directed dependencies among cluster, i.e., cluster i is executed before cluster j if there is a direct edge from cluster i to j . (3) There are two possible cases when an instruction needs data: (i) whenever it loads data from memory and (ii) whenever it depends on another instruction. When it needs data from memory, it can be in any cluster. For example, when an instruction from cluster i needs data from another instruction that is in cluster j , the topological sort during mapping stage resolves this situation by asking cluster i to wait before the completion of cluster j to make sure the data is available and sent to cluster i . To prevent livelock and deadlock situations, the optimization model used in partitioning has a constraint that prevents cyclic dependencies in clusters.

[1] (Continued) **In either case, I would expect a discussion of the limitations of the approach and examples of code it won't be able to partition.**

Thank you very much for your suggestion. As recommended, we discuss the limitations of current approach and a few directions for future work of the PGL, which are now added in the Discussion Section.

Figure 2 illustrates some of the limitations of our current approach. We summarize this limitation on page 15-16 in the revised manuscript.

- First, the runtime profiling in the PGL only supports C and C++ code that involves complicated computation. Simple code with only a few lines is not beneficial in PGL (see Figure 2(a) for a simple illustration example where a kernel contains three operations and it only has two clusters where one contains one instruction and the other contains two instructions after partitioning. The small code is not ideal in PGL because the overhead it spends on profiling and partitioning is not mitigated by mapping only a few instructions onto a specific core. In the future, developers could build more runtime systems that support different languages.
- Second, the PGL is not suitable for high level programs that involves many memory random accesses, due to memory dependencies that are hard to identify (see Figure 2(b)). As shown in Figure 2(b), we have a dynamic execution graph that involves memory address manipulation and indexing, which could lead to many false memory dependencies between

(a) Few instructions -> not beneficial to parallelize

(b) Pointer manipulation -> random memory access -> false memory dependencies in alias analysis
-> may increase the communication overhead

Figure 2: PGL Limitations. (a). It is not beneficial for small code to pay for the overhead of PGL while mapping a few instructions onto cores. (b). Random memory accesses from pointer manipulation are not beneficial in PGL because there will be thousands of false memory dependencies due to LLVM alias analysis. This may increase the communication overhead.

clusters. While the LLVM alias analysis reports the MustAlias, MayAlias, and NoAlias dependencies, we treat MustAlias and MayAlias as memory dependencies and add an edge between two instructions irrespective of whether they are must or may alias. This may increase the communication overhead due to too many MayAlias cases. On one hand, after we partition the dynamic execution graph into clusters to minimize inter-cluster communication, if most of the memory dependencies are confined in one cluster, then it would not increase communication. On the other hand, if many false memory dependencies span across different clusters, then communication overhead gets worse. In the future, this can be considered in the optimization model to partition the graph.

- Third, the high level programs have to be compiled and run successfully to collect the execution trace that is required to build the dynamic execution graph, which could be time and space consuming. In the future, developers could mitigate this issue by combing code runtime profiling and graph construction on the fly.

We followed the reviewer’s suggestion and stated these limitations and correctness issue in the revised manuscript as well as adding discussion for research opportunities to be addressed by future work that could improve PGL.

[2] It’s great that you acknowledge some work in this area. The second-newest paper you cite there is 29 years old, I’d hardly call that ”state of the art”. In particular, the point of this comment is not to check a box nor have you cite a specific paper, it’s to understand how your approach builds on and or extends existing work and frame it in that context. You say that ”The target platform in [SSKH13] is multi-core systems whereas we focus on the heterogeneous system that consists of GPUs and CPUs”. The former subsumes the latter: there’s a significant amount of work of mapping to heterogeneous multi-cores, with DSPs, GPUs, and other accelerators. It’s true that there’s a difference between on-chip latencies and off-chip heterogeneous systems that you mention, but those boundaries are blurring (see e.g. NVIDIA’s newly announced ”grace hopper” architecture) To be frank, I don’t particularly like the SSKH13 paper, it was just a reference that is somewhat comprehensive, there’s been much work since.

Thank you very much for your comment and suggestion. Indeed, there is a significant amount of work on mapping different models of computation to various computing systems that would be beyond the scope of this paper. We followed your suggestion of enhancing the prior work discussion and provide an updated discussion on the state-of-the-art graph representations of the high level programs and their corresponding runtime mapping in the revised manuscript on page 3.

The scheduling and mapping of dataflow graphs are a well-studied research area including synchronous dataflow [lee87, LM87] and dynamic dataflow [buc93]. [SL93a, SL93b] extend the job-shop scheduling techniques to account for interprocessor communication costs. Pino et al. [PPL94] show how to construct schedules for heterogeneous multiprocessors. Falk et al. [FKH⁺08] give a parallel scheduling strategy based on clustering and demonstrate significant performance gains for multimedia applications. In recent work [WO18], most approaches to deep learning in compiler optimization borrow ideas from deep learning in natural language processing. However, compiler domain has identified data structures such as abstract syntax tree and dataflow that exhibit the aspects more important for compiler optimization than the token sequences in natural language processing. Therefore, new graph representations of source code are developed to be used with the help of the recent advances in graph-based deep learning models such as graph neural networks. For example, [BGEC20] proposed a new compiler-based graph representation for deep learning models of code. It incorporates the abstract syntax tree and control-data-flow graphs to understand program properties and enable deep learning such as graph neural networks on the graph properties. In addition, the concurrent execution of varying mixes of different applications on the many-core systems enables state-of-the-art research in predictable application execution in terms of runtime mapping. For example, [WGW⁺14] proposed a hybrid mapping that achieves run-time predictability by combining the design-time analysis of application mappings with run-time management. [SWT22] provided a general, completely automated hybrid application mapping methodology for optimizing the mappings of multiple concurrent running soft real-time applications to a heterogeneous multiprocessor system on a chip to minimize latency and energy. However, previous work on graph representation of code fails to expose some interesting graph motifs in programming languages that are recurring at different scales. The proposed dynamic execution graph illustrates different self-repeating code structures that can be exploited in multifractal analysis to extract meaningful features.

However, we would like to point out that the heterogeneous device mapping problem is somewhat different from the traditional mapping in computer architecture and system design. The heterogeneous device mapping problem is to predict the device (CPU or GPU) which will provide the best performance, given a kernel. It first originated from [GWO13] where the framework uses predictive modeling to automatically determine if it is beneficial to run the OpenCL code on the GPU or OpenMP code on the multi-core host. The problem is then further studied and analyzed using machine learning models [BNJH18, CPWL17]. Once a device is predicted, we are still relying on the compiler and runtime system to map the kernel to the actual hardware. It does not replace traditional mapping. Instead, it offers the runtime system with a choice to choose a better type of hardware for higher performance.

- [2] (Continued) **The point you make about the granularity is a very valid one and should probably be in the paper (more so than, say, enumerating some selection of 30-year-old papers). It's interesting that you want to use a finer granularity for systems with a higher communication overhead; how can you get away with this, compared to the state of the art?**

Thank you very much for your comment. The motivation for adopting a finer granularity analysis is three-fold: Firstly, the high level languages and high level programs may be designed in order to optimize certain software engineering objectives (e.g., modularity), but they are not taking advantage of or keep up with recent hardware innovations and developments (e.g., high parallelism in exascale computing). Secondly, the software development for certain applications may be done in a suboptimal way without considering the time complexity in algorithms such as recursion used in Fibonacci numbers that leads to $O(2^N)$ where N is the N th Fibonacci number. Thirdly, to bridge the gap between the high performance offered by heterogeneous hardware platforms and high flexibility offered by general purpose computing, we need a model

Table 1: Dataset summary

	Description	Source	Number of kernels
1	Algebraic multigrid solver (AMS)	[amd]	8
2	Fast sequence alignment (FSA)	[LTPS09, bow]	20
3	DNA sequence mapping (DSM)	[Li13, bwa]	21
4	Neural network (NN)	[nn]	2
5	Dijkstra algorithm (DA)	[dij]	1
6	Epidemic simulation (ES)	[FLNG ⁺ 20, epi]	1
7	Molecular dynamics (MD)	[md]	2
8	Graph partitioning (GP)	[DA97, par]	17
9	Euler equation solver (EES)	[ees]	9
10	Evolutionary algorithm (EA)	[KDW22, evo]	4
11	IO proxy application (IPA)	[iop]	6
12	Mesh refinement application (MRA)	[amr]	16
13	Convolutional neural network (CNN)	[FYP ⁺ 21, cnn]	6
14	Poisson equation solver (PES)	[GML ⁺ 16, nek]	3
15	Monte Carlo kernel (MCK)	[xsb]	3
16	HACC	[hac]	9
17	Radiative transfer solver (RTS)	[ime]	4

of computation representation that allows us to flexibly capture the best of both worlds - the software and the hardware. Towards this end, we adopted the dynamic execution graphs with a finer-grain assembly code representation to retain the above-mentioned flexibility and provide higher software-hardware flexibility when compared to the dataflow graphs used in the literature. However, this finer granularity does not necessarily mean higher communication overhead. Higher granularity means more nodes and more edges in our implementation but the communication overhead refers to the amount of communication that takes place between clusters after the partitioning. In order to prevent higher communication overhead, we introduce our partitioning algorithm that partitions the dynamic execution graphs into clusters. Indeed, we expect that a resulting cluster from the partitioning operation to be more similar to a node in the dataflow graphs in the literature. Each cluster is a sequence of instructions that are optimized to reduce data communication between clusters. Therefore, our graph representation with partitioning does not have a higher communication overhead. We optimize the inter-cluster communication to make sure the communication overhead between clusters is minimized.

- [3] **Thanks for addressing the communication overhead. My question was not how do you model it, it was how big is it, what percentage of the execution time does it represent? I would recommend removing this discussion on how you model it, and instead, measuring it and discussing how large the overhead was.**

Thank you very much for the suggestion and we apologize for the misunderstanding. We followed your recommendation and we measured the communication overhead as follows: We performed experiments on two different benchmark suites in terms of application performance and reported the clock cycles spent either on communication or computation as shown in Figure 3 and 4. For example, memory-intensive applications such as Dijkstra involves pointer address manipulation that requires frequent data fetch from memory. When running in PAR without any optimization, the communication overhead compared to the execution time is 820.52×10^7 . However, PGL manages to reduce it to 109.38×10^7 via the optimization model to partition the graph into clusters while minimizing inter-cluster communication. For compute-intensive applications such as FFT, the communication overhead running in PAR is 63.18×10^7 whereas it is reduced to 31.10×10^7 in PGL. This indicates that even though PGL uses a finer-grained graph representation, the optimization based partitioning approach would reduce the communication overhead between clusters.

- [4] **I frankly did not understand the response. Why as input to a neural network? Why can they not be used in your dataset? More concretely: A part of your system estimates the performance of a kernel by using dynamic information. My question is "how accurate is this?", compared to e.g. LLVM-MCA or other more sophisticated**

Figure 3: The breakdown of the execution time of each application in the standard dataset running on different frameworks. The execution time, measured in clock cycles, is roughly divided into two parts: communication and computation in (a). We also report communication overhead that is calculated by clock cycles in communication divided by the total clock cycles in (b). As we can see, PGL, compared to the other frameworks, has the smallest communication overhead. It is because PGL has an optimization model that partitions the graph into different clusters to minimize inter-cluster communication.

Figure 4: The breakdown of the execution time of each application in the real-life dataset in Table 1 running on different frameworks. The execution time, measured in clock cycles, is roughly divided into two parts: communication and computation in (a). We also report communication overhead that is calculated by clock cycles in communication divided by the total clock cycles in (b).

Table 2: Configuration parameters

CPU	Cores	64 In-order cores, 16 MSHRs
	Clock frequency	2.4 GHz
	L1 private cache	128KB, 4-way associative 32-byte blocks
	L2 shared cache	2MB, distributed
	Memory	4 GB, 8 GB/s bandwidth
GPU	Number	512
	Clock Frequency	575 MHz
	Memory	768 MB, 86.4 GB/s bandwidth
Network	Topology	Mesh
	Routing algorithm	XY routing
	Flow control	Virtual channel flit-based

approaches. In other words, if you split up only your performance estimation, how much better does it estimate the performance of a piece of code? I'm not asking you to replace that part of your system with LLVM-MCA, I want to understand how good that part of the system is, compared to a standard baseline.

We apologize for the confusion. We do not estimate the performance of a kernel. More precisely, given a kernel and a choice of two devices to run it on (CPU or GPU), our machine learning framework is trained to learn a model that can predict the device (CPU or GPU) which will provide the best performance, without estimating the kernel performance. The training of the machine learning model requires input kernels and labels. Each input in our framework is a graph for a kernel and a label is either CPU or GPU, which can be encoded as 0 and 1. A label is decided by comparing the kernel runtimes on the CPU vs the GPU, which is collected from real hardware. We are not using a performance estimator for it. During evaluation, we use our framework to predict the device (CPU or GPU), and we use gem5 (a computer architecture simulator) to estimate the performance.

- [5] **Thanks for the clarification. I know what LLVM and SSA are, and I think calling it a universal IR is controversial at best; this change clarifies and addresses my doubt nevertheless, it was not clear to me that you meant LLVM by that.**

Thank you very much for your feedback.

- [6] **Fair enough. The example is only somewhat convincing, as the Strassen algorithm is over 50 years old and was the state of the art until last year's AlphaZero result. Nevertheless, I get the spirit of the answer and I'm happy to accept this point.**

Thank you very much for your feedback.

- [7-12] **Great!**

Thanks for the feedback.

- [13] **This is a significant improvement and insightful, thanks.**

Thanks for the feedback.

- [14] **The explanation seems mathematically reasonable, although I'm not familiar with this topic. I think it's still not very insightful though. Computer engineers will not gain much from the mathematics there, whereas a geometer who studies fractals will not understand much about the compiler side. What's your target audience? I think if you're describing the application, you can focus on how the concepts are useful and intuitively explain what's happening, while safely referring to literature for the formal definitions and technical details. The goal of this is not to criticize your work, but rather, to suggest ways in which it can be made more accessible to your target audience, as I understand it.**

Thank you very much for your suggestion. We followed your suggestion and have moved the discussion of multifractal analysis to supplementary materials. Instead, we are discussing the

high-level picture of the analysis and how it can help us understand the structure of the code, particularly how can we capture the long-range dependencies that may exist in a program.

Dynamic execution graphs can exhibit some self-repeating patterns on different scales that we can exploit to capture and understand the intrinsic graph structures. There are two fundamental techniques in programming languages: iteration and recursion. Iteration is a significant routine for a program to define a number of repetitions, usually via for-loops and while-loops. It corresponds to the mesh-like topology in graph representation. Recursion is the other major approach for a program to solve a problem where the solution depends on solutions to smaller instances of the same problem. It corresponds to a tree-like topology in graph representation. These two types of graphs can be analyzed at different scales to understand the recurring structures to extract the hidden features. For example, Figure 5 shows an example of a for loop and its corresponding graph structure. In order to analyze its self-repeating patterns that can be seen in (b), we use the box-counting algorithm in multifractal analysis to calculate the dominant fractal dimension. In other words, we follow the definition of the measure to find the number of boxes ($N(B)$) with a box size r . The number of boxes is calculated by the optimal amount used to cover the entire graph. For example, when $r = 1$, the number of boxes $N(B)$ is the number of nodes in the graph, which is 116 in this case. When r is the diameter of the graph, the number of boxes $N(B)$ is 1.

Once we analyze different graph properties from multifractal analysis such as generalized fractal dimension, spectrum height and width, we are trying to relate the graph-level properties with some system-level metrics such as communication overhead and parallelization degree by fitting a power-law model into the data to help us understand the relationship. As we can see in Figure 6, There exists such model that can approximately estimate the system-level metrics from graph properties. This has two folds.

1. It provides a universal model that builds the relationship between the graph properties such as the multifractal spectrum and the system-level metrics such as the parallelization degree and communication overhead. If such a model can accurately capture the relationship, the optimal degree of parallelization can be calculated by the graph properties, without the manual tuning from a programmer. For example, if a dynamic execution graph from a piece of code has a spectrum width that equals 2, then we would expect communication overhead between 80 and 120 *times* 10^8 clock cycles, and the parallelization degree between 10 and 24. On the other hand, if a future platform can support millions of cores, then we can use this model to find how to write the code that can exploit the benefits in exascale computing.
2. It provides us with what design choices we can gain to develop the feature extraction algorithm. PGL contains a feature extraction algorithm used by the graph neural network to predict a label. It shows that multifractal analysis can capture the graph topological structures, which can be further used in the feature extraction algorithm.

[15-16] **Great!**

Thanks for the feedback.

[17] **The comparison to state of the art is confusing: how is gem5-aladdin, a simulator, a comparison to PGL? More importantly, how is this, and a generic neural network with reinforcement learning state-of-the-art? It certainly is not state of the art in mapping (see above), which explains how the numbers of the speedup can be so high!**

We apologize for the confusion. We believe this confusion can be mitigated by changing gem5-aladdin to Aladdin. Explicitly mentioning gem5 in the framework is confusing because it creates an illusion that only gem5-aladdin is using gem5. In fact, all of the frameworks in the performance evaluation, i.e., PAR, CommDet, Aladdin, NN+RL, and PGL, are using gem5 simulation environment to evaluate the performance of each approach for a fair comparison.

1. PAR: The code is written in OpenCL or OpenMP to achieve parallel processing.

Figure 5: Example of code with its graph representation and box counting algorithm used to analyze the multifractal properties. (a) A loop kernel; (b) The dynamic execution graph with a zoom-in view on one iteration of the loop; (c) The box counting algorithm by varying the size of a box r to count the number of boxes $N(B)$.

Figure 6: Multifractal analysis can characterize the universal power-law relationship between multifractal properties and system-level metrics. Network multifractal properties are used as inputs to fit a power-law model ax^b to find the relationship between network properties and system-level metrics. Figure (a-f) shows the parallelization degree of code graphs in terms of generalized fractal dimension (a-b), spectrum width (c), spectrum height (d), α_0 (e), and complexity (f). Figure (g-i) shows the communication overhead for spectrum width (g), α_0 (h), and complexity (i).

Figure 7: (a). Comparison of different partitioning algorithms. We compare the graph partitioning GAE with different traditional algorithms. (b). Comparison of different frameworks. We compare PGL with different frameworks in terms of application performance. We conclude that our approach can achieve 2.02x better compared to the state-of-the-art techniques.

2. CommDet: The graph is partitioned via the community detection algorithm to reduce communication overhead and balance workloads [XXNB17].
3. Aladdin: It enables rapid design space search of accelerator-centric systems by taking dynamic data dependence graphs from an application as a representation of an accelerator to create a realistic model of accelerator activity [SRWB14].
4. NN+RL: The graph is analyzed by neural networks to identify some common graph motifs that can be accelerated on hardware, which are further mapped with the help of reinforcement learning [XNB19].

We provide the details about the gem5 configuration in Table 2. More specifically, Aladdin [SRWB14] explores the large design space to choose a design point. Therefore, we are comparing PGL to Aladdin (or gem5-PGL to gem5-aladdin) to validate the effectiveness of PGL. You can find the modified Figure 7 where we replace gem5-aladdin with Aladdin for a clear understanding of the evaluation.

We also compare three different graph representations in order to validate the effectiveness of PGL: dynamic data dependence graphs (DDDg) in Aladdin [SRWB14], control-flow / data-flow / call-flow graphs (CDCFG), and dynamic execution graphs (DEG). In a DDDg, each node represents computation and each edge represents dynamic data dependence between nodes. However, compared to DEG, it fails to capture the memory dependencies that would be the main bottleneck in some memory-intensive applications. Each CDCFG is constructed in such a way that combines control-flow graphs (CFG), data-flow graphs (DFG), and call-flow graphs (CFG). We evaluate the same datasets on these three graph representations and measure the application performance in terms of clock cycles, as shown in Figure 8. We found out that DEG, compared to DDDg and CDCFG, provides on average 1.69x improvement and is suitable for both compute-intensive and memory-intensive applications. This discussion can be found in page 7-8 in the revised manuscript.

References

- [amd] Algebraic multigrid benchmark. <https://github.com/LLNL/AMG>.
- [amr] MiniAMR Adaptive Mesh Refinement (AMR) Mini-App. <https://github.com/Mantevo/miniAMR>.
- [BGEC20] Alexander Brauckmann, Andrés Goens, Sebastian Ertel, and Jeronimo Castrillon. Compiler-based graph representations for deep learning models of code. In *Proceedings of the 29th International Conference on Compiler Construction*, pages 201–211, 2020.
- [BNJH18] Tal Ben-Nun, Alice Shoshana Jakobovits, and Torsten Hoefler. Neural code comprehension: A learnable representation of code semantics. *arXiv preprint arXiv:1806.07336*, 2018.

Figure 8: Application performance in the (a) standard dataset and (b) real-life dataset running on different graph representations.

- [bow] Fast and sensitive gapped read aligner. <https://github.com/BenLangmead/bowtie2>.
- [buc93] Scheduling dynamic dataflow graphs with bounded memory using the token flow model. In *1993 IEEE international conference on acoustics, speech, and signal processing*, volume 1, pages 429–432. IEEE, 1993.
- [bwa] Burrow-Wheeler Aligner for short-read alignment. <https://github.com/lh3/bwa>.
- [cnn] CNN in face detection. <https://github.com/ShiqiYu/libfacedetection>.
- [CPWL17] Chris Cummins, Pavlos Petoumenos, Zheng Wang, and Hugh Leather. End-to-end deep learning of optimization heuristics. In *2017 26th International Conference on Parallel Architectures and Compilation Techniques (PACT)*, pages 219–232. IEEE, 2017.
- [DA97] Ali Dasdan and Cevdet Aykanat. Two novel multiway circuit partitioning algorithms using relaxed locking. *IEEE Transactions on Computer-Aided Design of Integrated Circuits and Systems*, 16(2):169–178, 1997.
- [dij] Dijkstra shortest path. <https://github.com/Lehmannhen/MPI-Dijkstra>.
- [ees] High-order Lagrangian Hydrodynamics Miniapp. <https://github.com/CEED/Laghos>.
- [epi] Epidemic simulation software. https://github.com/ZiluTian/epidemic_simulation.
- [evo] Lamarckian evolutionary algorithm for de novo drug design. <https://github.com/UAMCAntwerpen/LEADD>.
- [FKH⁺08] Joachim Falk, Joachim Keinert, Christian Haubelt, Jürgen Teich, and Shuvra S Bhat-tacharyya. A generalized static data flow clustering algorithm for mpsoc scheduling of multimedia applications. In *Proceedings of the 8th ACM international conference on Embedded software*, pages 189–198, 2008.
- [FLNG⁺20] Neil M Ferguson, Daniel Laydon, Gemma Nedjati-Gilani, Natsuko Imai, Kylie Ainslie, Marc Baguelin, Sangeeta Bhatia, Adhiratha Boonyasiri, Zulma Cucunubá, Gina Cuomo-Dannenburg, et al. Impact of non-pharmaceutical interventions (npis) to reduce covid-19 mortality and healthcare demand. 2020.
- [FYP⁺21] Yuantao Feng, Shiqi Yu, Hanyang Peng, Yan-Ran Li, and Jianguo Zhang. Detect faces efficiently: A survey and evaluations. *IEEE Transactions on Biometrics, Behavior, and Identity Science*, 4(1):1–18, 2021.
- [GML⁺16] Jing Gong, Stefano Markidis, Erwin Laure, Matthew Otten, Paul Fischer, and Misun Min. Nekbone performance on gpus with openacc and cuda fortran implementations. *The Journal of Supercomputing*, 72(11):4160–4180, 2016.
- [GWO13] Dominik Grewe, Zheng Wang, and Michael FP O’Boyle. Portable mapping of data parallel programs to opencl for heterogeneous systems. In *Proceedings of the 2013 IEEE/ACM International Symposium on Code Generation and Optimization (CGO)*, pages 1–10. IEEE, 2013.
- [hac] Stand-alone version of HACC’s distributed-memory, pencil-decomposed, parallel 3D FFT. <https://xgitlab.cels.anl.gov/hacc/SWFFT>.
- [ime] IMEX transport mini-app. https://github.com/ECP-Astro/thornado_mini.
- [iop] Multi-purpose, Application-Centric, Scalable I/O Proxy Application. <https://github.com/LLNL/MACsio>.
- [KDW22] Alan Kerstjens and Hans De Winter. Leadd: Lamarckian evolutionary algorithm for de novo drug design. *Journal of cheminformatics*, 14(1):1–20, 2022.
- [lee87] Synchronous data flow. *Proceedings of the IEEE*, 75(9):1235–1245, 1987.

- [Li13] Heng Li. Aligning sequence reads, clone sequences and assembly contigs with bwa-mem. *arXiv preprint arXiv:1303.3997*, 2013.
- [LM87] Edward Ashford Lee and David G Messerschmitt. Static scheduling of synchronous data flow programs for digital signal processing. *IEEE Transactions on computers*, 100(1):24–35, 1987.
- [LTPS09] Ben Langmead, Cole Trapnell, Mihai Pop, and Steven L Salzberg. Ultrafast and memory-efficient alignment of short dna sequences to the human genome. *Genome biology*, 10(3):1–10, 2009.
- [md] Molecular dynamics proxy application. <https://github.com/ECP-copa/ExaMiniMD>.
- [nek] Nekbone software. <https://github.com/AMDCComputeLibraries/Nekbone>.
- [nn] Open source neural network. <https://github.com/QuentinAM/neural-network-c>.
- [par] Multi-way graph partitioning algorithms. <https://github.com/alidasdan/graph-partitioning-algorithms>.
- [PPL94] Jose Luis Pino, Thomas M Parks, and Edward A Lee. Automatic code generation for heterogeneous multiprocessors. In *Proceedings of ICASSP’94. IEEE International Conference on Acoustics, Speech and Signal Processing*, volume 2, pages II–445. IEEE, 1994.
- [SL93a] Gilbert C Sih and Edward A Lee. A compile-time scheduling heuristic for interconnection-constrained heterogeneous processor architectures. *IEEE transactions on Parallel and Distributed systems*, 4(2):175–187, 1993.
- [SL93b] Gilbert C Sih and Edward A Lee. Declustering: A new multiprocessor scheduling technique. *IEEE Transactions on Parallel and Distributed Systems*, 4(6):625–637, 1993.
- [SRWB14] Yakun Sophia Shao, Brandon Reagen, Gu-Yeon Wei, and David Brooks. Aladdin: A pre-rtl, power-performance accelerator simulator enabling large design space exploration of customized architectures. *ACM SIGARCH Computer Architecture News*, 42(3):97–108, 2014.
- [SWT22] Jan Spieck, Stefan Wildermann, and Jürgen Teich. A learning-based methodology for scenario-aware mapping of soft real-time applications onto heterogeneous mpsocs. *ACM Transactions on Design Automation of Electronic Systems*, 28(1):1–40, 2022.
- [WGW⁺14] Andreas Weichslgartner, Deepak Gangadharan, Stefan Wildermann, Michael Glaß, and Jürgen Teich. Daarm: Design-time application analysis and run-time mapping for predictable execution in many-core systems. In *Proceedings of the 2014 International Conference on Hardware/Software Codesign and System Synthesis*, pages 1–10, 2014.
- [WO18] Zheng Wang and Michael O’Boyle. Machine learning in compiler optimization. *Proceedings of the IEEE*, 106(11):1879–1901, 2018.
- [XNB19] Yao Xiao, Shahin Nazarian, and Paul Bogdan. Self-optimizing and self-programming computing systems: A combined compiler, complex networks, and machine learning approach. *IEEE Transactions on Very Large Scale Integration (VLSI) Systems*, 27(6):1416–1427, 2019.
- [xsb] XSBench: The Monte Carlo Macroscopic Cross Section Lookup Benchmark. <https://github.com/ANL-CESAR/XSBench>.
- [XXNB17] Yao Xiao, Yuankun Xue, Shahin Nazarian, and Paul Bogdan. A load balancing inspired optimization framework for exascale multicore systems: A complex networks approach. In *2017 IEEE/ACM International Conference on Computer-Aided Design (ICCAD)*, pages 217–224. IEEE, 2017.

REVIEWERS' COMMENTS:

Reviewer #2 (Remarks to the Author):

Dear Authors,

Please apologize for the late response, I wanted to have a proper look at the changes and the last few weeks were chaotic to find time.

I've reviewed the changes to the paper and your responses. I think the paper has improved significantly, in its scope and content. Particularly the new results about the overheads are very enlightening, and many of the clarifications e.g. about the comparison resolve the main issues I pointed out, as well as the new explanation on the multi-fractal methods (and moving the previous one to supplementary materials), I think these made the paper more appropriate for the target audience.

Thank you for addressing the changes! I'm happy to recommend this for publication!

Response to Referees

August 17, 2023

1 Reviewer #2

Dear Authors, Please apologize for the late response, I wanted to have a proper look at the changes and the last few weeks were chaotic to find time. I've reviewed the changes to the paper and your responses. I think the paper has improved significantly, in its scope and content. Particularly the new results about the overheads are very enlightning, and many of the clarifications e.g. about the comparison resolve the main issues I pointed out, as well as the new explanation on the multi-fractal methods (and moving the previous one to supplementary materials), I think these made the paper more appropriate for the target audience. Thank you for addressing the changes! I'm happy to recommend this for publication!

Authors: We thank the second reviewer for his feedback and suggestions for improvement.